# A TrkB agonist prodrug prevents bone loss via inhibiting asparagine endopeptidase and increasing osteoprotegerin

Jing Xiong [1,2,6], Jianming Liao [1,3,6], Xia Liu[1], Zhaohui Zhang[2], Jonathan Adams[4], Roberto Pacifici [4] & Keqiang Ye [1,5] ✉

Brain-derived neurotrophic factor (BDNF) and its tropomyosin-related kinase B receptor (TrkB) are expressed in human osteoblasts and mediate fracture healing. BDNF/TrkB signaling activates Akt that phosphorylates and inhibits asparagine endopeptidase (AEP), which regulates the differentiation fate of human bone marrow stromal cells (hBMSC) and is altered in postmenopausal osteoporosis. Here we show that R13, a small molecular TrkB receptor agonist prodrug, inhibits AEP and promotes bone formation. Though both receptor activator of nuclear factor kappa-B ligand (RANK-L) and osteoprotegerin (OPG) induced by ovariectomy (OVX) remain comparable between WT and BDNF+/− mice, R13 treatment significantly elevates OPG in both mice without altering RANKL, blocking trabecular bone loss. Strikingly, both R13 and anti-RANK-L exhibit equivalent therapeutic efficacy. Moreover, OVX increases RANK-L and OPG in WT and AEP KO mice with RANK-L/OPG ratio lower in the latter than the former, attenuating bone turnover. 7,8-DHF, released from R13, activates TrkB and its downstream effector CREB, which is critical for OPG augmentation. Consequently, 7,8-DHF represses C/EBPβ/AEP pathway, inhibiting RANK-L-induced RAW264.7 osteoclastogenesis. Therefore, our findings support that R13 exerts its therapeutic efficacy toward osteoporosis via inhibiting AEP and escalating OPG.

Brain-derived neurotrophic factor (BDNF) belongs to the family of neurotrophins that play essential roles in the central nervous system (CNS) and are mainly expressed in central and peripheral neuronal tissues[1,2]. However, BNDF is also synthesized and released from non-neuronal cells such as fibroblasts, osteoblasts, endothelial cells, monocytes, and mast cells[3,4]. Plasma BDNF levels are increased in patients with osteoarthritis compared to healthy individuals[5]. BDNF is involved in osteoblast cell differentiation and stimulates bone/cementum-related proteins including alkaline phosphatase (ALP),

bone morphogenetic protein-2, and osteopontin expression in cementoblasts[6]. Both BDNF and its TrkB receptor are present at various stages of the bone formation process, and they are upregulated in human osteoblasts and implicated in fracture healing[7]. BDNF strongly elevates mRNA expression of the osteoblast differentiation marker, osteocalcin, in the osteoblast-lineage cell MC3T3-E1 and stimulates cell differentiation and promotes new bone formation and maturation[8].

Asparaginyl endopeptidase (AEP, also known as legumain with gene name: LGMN) is a broadly expressed endo-lysosomal cysteine

[1]Department of Pathology and Laboratory Medicine, Emory University School of Medicine, Atlanta, GA 30322, USA. [2]Department of Neurology, Renmin Hospital of Wuhan University, Wuhan 430060 Hubei Province, PR China. [3]Department of Neurosurgery, Renmin Hospital of Wuhan University, Wuhan 430060 Hubei Province, PR China. [4]Division of Endocrinology, Metabolism and Lipids, Department of Medicine, Emory University School of Medicine, Atlanta, GA 30322, USA. [5]Faculty of Life and Health Sciences, Shenzhen Institute of Advanced Technology (SIAT) Shenzhen, Guangdong, PR China. [6]These authors contributed equally: Jing Xiong, Jianming Liao. ✉e-mail: kq.ye@siat.ac.cn

protease that is secreted as inactive pro-zymogen (56 kDa) and processed into an enzymatically active 36 kDa mature form and a 11 kDa C-terminal inhibitory fragment[9]. Strikingly, the C-terminal truncate inhibits osteoclast differentiation through binding to an uncharacterized receptor[10,11]. Active AEP inhibits osteoblast differentiation and in vivo bone formation through degradation of the bone matrix protein, fibronectin. During development, AEP-deficient zebrafish exhibits precocious bone formation and mineralization[12]. Human bone marrow stromal cells (hBMSCs) are non-hematopoietic multipotent cells capable of differentiation into mesodermal cell types such as osteoblasts and adipocytes[13]. Markedly, AEP regulates the lineage commitment of hBMSCs and is abnormally expressed and displays aberrant subcellular localization in the bone from patients with postmenopausal osteoporosis[12].

We have described 7,8-dihydroxyflavone (7,8-DHF) as a small molecule that mimics BDNF and acts as a specific TrkB agonist with high binding affinity. After binding to the extracellular motif on TrkB receptor, 7,8-DHF triggers receptor dimerization and auto-phosphorylation, initiating neurotrophic activities[14–16]. It is well documented that 7,8-DHF simulates BDNF biologic functions and exerts promising therapeutic efficacy toward a variety of diseases implicated with BDNF/TrkB signaling[17–20]. To improve its in vivo pharmacokinetic (PK) profiles, we have prepared a prodrug, R13, that releases 7,8-DHF after absorption and significantly increases its oral bioavailability[21–23]. Recently, we reported that BDNF/TrkB signaling inhibits AEP via Akt phosphorylation of the T322 residue, suppressing AEP activation[24]. Oral administration of R13 elicits robust TrkB receptor activation in the brain and the gut and inhibits AEP via Akt-mediated T322 phosphorylation[21]. Moreover, C/EBPβ is a pivotal transcription factor for escalating AEP expression during age[25], and activation of the BDNF/TrkB pathway represses C/EBPβ/AEP signaling[26].

Osteoporosis is a systemic bone disease, characterized by reduced bone mass, and disruption of normal bone architecture, resulting in bone fragility and increased risk of fractures[27]. Bone homeostasis depends on the resorption of bones by osteoclasts and formation of bones by the osteoblasts. Osteoblasts can also affect osteoclast formation, differentiation, or apoptosis through several pathways, such as OPG/RANK-L/RANK. In the current study, to test the hypothesis that the BDNF mimetic drug R13 may block AEP and promote bone formation, we employed BDNF+/−, AEP −/−, and wild-type (WT) littermate mice and examined their roles in ovariectomy (OVX)-induced bone loss in the presence or absence of R13. We found that AEP KO decreased OVX-induced bone loss via increasing osteoblast formation and inhibiting osteoclast formation. 7,8-DHF, the active pharmaceutical ingredient released from R13, elevates OPG expression via activating CREB and blocks RANK-L-induced osteoclastogenesis. R13 not only represses AEP expression through blunting its upstream transcription factor C/EBPβ but also blocks AEP activation via BDNF/TrkB pathway-activated Akt and it displays promising therapeutic efficacy toward osteoporosis which is similar to anti-RANKL antibody.

## Results

### Knockout of AEP improves trabecular bone density in ovariectomized female mice

To explore the role of AEP in bone remodeling, we subjected AEP knockout mice (AEP KO) and WT littermates to OVX at the age of 12 weeks. As expected, the shrunken uterine morphology and reduced uterus weight revealed that OVX surgery was successful (Supplementary Fig. 1). Microcomputed tomography (μCT) analysis of femurs harvested at sacrifice revealed a higher trabecular bone volume fraction (BV/TV), Conn.D and a lower Structure model index (SMI) in AEP KO mice compared with AEP WT mice after OVX. Moreover, OVX decreased trabecular number (Tb.N) and increased trabecular separation (Tb.Sp), while trabecular thickness (Tb.Th) indices were

similar among the groups. These indices remained similar between two types of mice under sham operation (Fig. 1a, b). μCT scanning demonstrated that cortical bone Cortical area (Ct.Ar) and average cortical thickness (Ct.Th) were reduced upon OVX in both WT and AEP KO mice, however, relative cortical bone area to tissue area (Ct.Ar/Tt.Ar) remained comparable among the groups (Fig. 1c). Notably, levels of serum osteocalcin, a marker of bone formation, were increased after OVX with AEP KO significantly higher than WT. The serum [BDNF]s were comparable among the four groups. Quantification of bone resorption indices in the serum showed that the concentrations of C-terminal telopeptide of collagen (CTX), a marker for bone resorption, and RANK-L were increased after OVX mice. Moreover, OPG concentrations were much higher in AEP KO mice than WT mice under both OVX and sham conditions, suggesting that AEP antagonizes OPG expression under the physiological condition. Consequently, the ratios of RANKL/OPG were substantially higher in OVX groups than sham groups with AEP KO mice lower than WT mice, in alignment with higher bone density in AEP KO group versus WT group after OVX (Fig. 1d). Hence, AEP deletion diminishes the ratio of RANKL/OPG, leading to increased trabecular bone density after OVX.

### Deletion of AEP inhibits the bone turnover induced by ovariectomy

To further characterize the roles of AEP in OVX-induced osteoporosis, we performed the H&E staining and analyzed the bone morphology and white adipocytes in both animals after OVX surgery. White adipocytes were evidently reduced and trabecular bone was increased in the bone from AEP KO mice after OVX as compared to WT mice (Fig. 2a). Tartrate-resistant acid phosphatase (TRAP) staining revealed that OVX induced more osteoclast cells in WT than AEP KO mice (Fig. 2b). Calcein double-fluorescence labeling allows the determination of the onset time and location of mineralization and the direction and speed of bone formation. Based on dynamic indices of femur trabecular bone formation, no significant difference in mineral apposition rate (MAR) and bone formation rate (BFR) was found between WT and AEP KO sham mice, but OVX significantly decreased the MAR and BFR in WT mice, as AEP knockout alleviated this difference (Fig. 2d). Representative data of double labeling in trabecular bone are shown in Fig. 2c. Analysis of static indices of bone formation and resorption revealed that both number of osteoclasts (N. Oc/BS) and the percentage of surfaces covered by osteoclasts (OcS/BS) were greatly decreased in AEP KO mice compared with WT mice after OVX. On the other hand, OVX also elicited a compensatory increase of number of osteoblasts (N. Ob/BS) in AEP WT group but not in AEP KO mice. Although the MS/BS (mineralizing surface/bone surface) ratios were significantly reduced in WT mice after OVX, they were unchanged in AEP KO mice (Fig. 2d). Together, these data suggest that AEP deficient mice exhibit a higher bone formation and lower bone resorption after OVX.

### R13 increases OPG level and blocks trabecular bone loss induced by ovariectomy

To explore the biological roles of BDNF/TrkB signaling in OVX-induced bone loss, we employed 12 weeks old female BDNF+/− mice and WT littermates. Four days after OVX surgery, WT and BDNF+/− mice were administered either R13 (21.8 mg/kg) or vehicle, orally, 6 days per week for 8 weeks. Assessment of femoral bone structure by in vitro μCT revealed that trabecular bone volume, expressed as a function of total tissue volume fraction (BV/TV), Conn.D was dramatically decreased, and SMI was increased by OVX in both WT and BDNF+/− mice. However, R13 treated-OVX mice showed a higher BV/TV, Conn.D and a lower SMI compared to OVX group. Quantification of parameters of trabecular structure revealed that R13-treated OVX mice displayed higher Tb.Th and Tb.N than vehicle control, decreased Tb.Sp in both type of mice as compared with the OVX-treated group (Fig. 3a, b). μCT

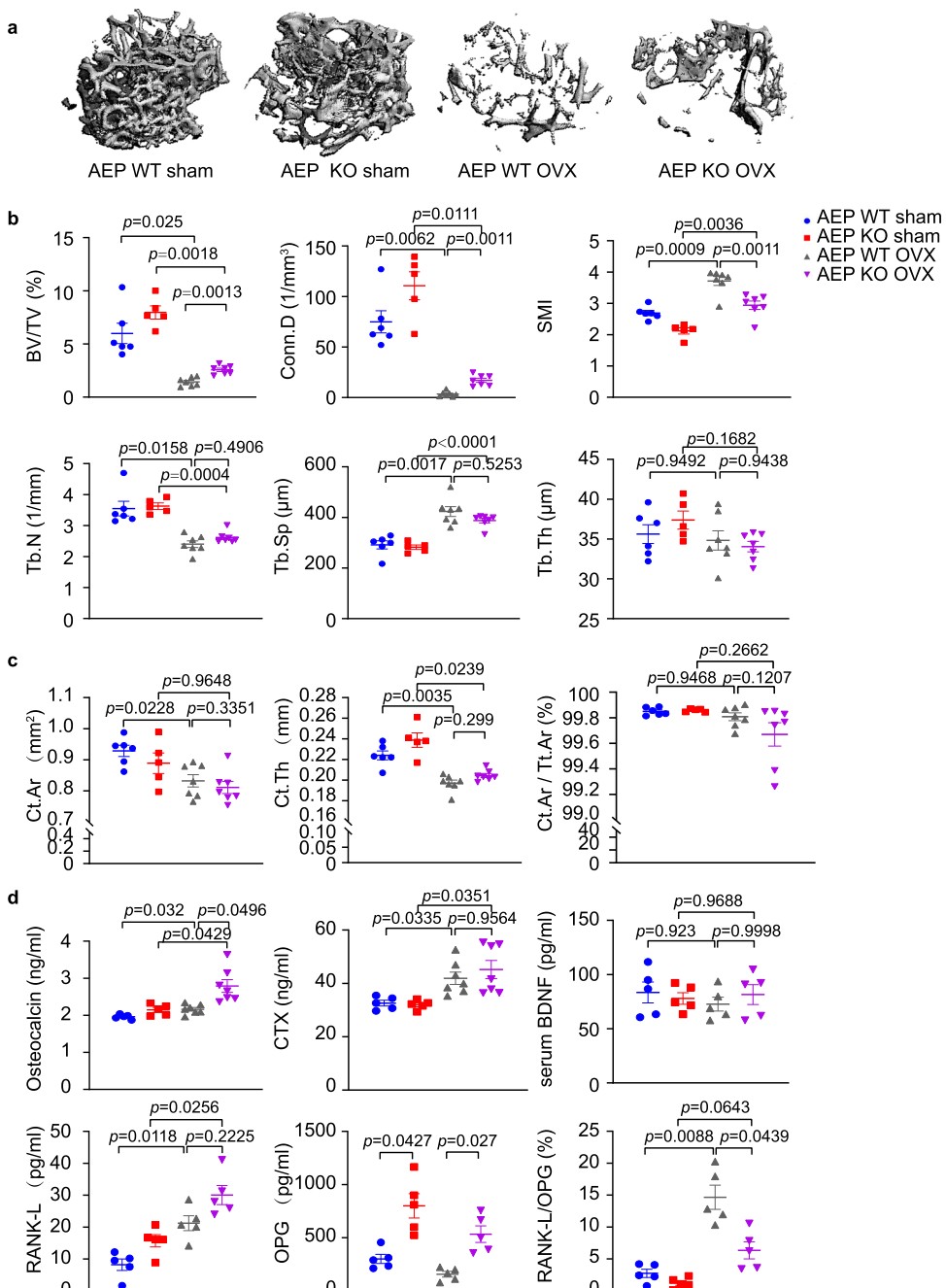

**Fig. 1 | AEP knockout improves trabecular bone density in ovariectomy female mice.** Femoral bone structures were assessed by in vitro μCT in AEP wild-type, AEP knockout (AEP KO) mice with or without ovariectomy. **a** Representative images of the femoral indices of trabecular bone structure measured by in vitro μCT scan. **b** μCT scanning measurements of trabecular bone volume fraction (BV/TV), connectivity density (Conn.D), structure model index (SMI), trabecular number (Tb.N), trabecular spacing (Tb.Sp), trabecular thickness (Tb.Th). Data are shown as mean ± SEM, $n = 5$ mice per group for AEP WT sham group, $n = 6$ mice per group for AEP KO group and $n = 7$ mice per group for AEP WT and AEP KO OVX group, one-way ANOVA. **c** μCT scanning measurements of cortical bone

cortical area (Ct.Ar), average cortical thickness (Ct.Th) and relative cortical bone area to tissue area (Ct.Ar/Tt.Ar). Data are shown as mean ± SEM, $n = 5$ mice per group for AEP WT sham goup, $n = 6$ mice per group for AEP KO group and $n = 7$ mice per group for AEP WT and AEP KO OVX group, one-way ANOVA. **d** Serum levels of osteocalcin (a marker of bone formation), C-terminal telopeptide (CTX) (a marker of bone resorption), RANK-L, OPG, RANK-L/OPG ratio and serum BDNF level in wild-type and AEP knock out mice with and without ovariectomy. Data are shown as mean ± SEM, left to right: $n = 5, 5, 7, 7$ mice per group for osteocalcein and CTX measurement, $n = 5$ mice per group for BDNF, RANK-L, and OPG measurement, one-way ANOVA.

scanning showed that cortical bone Cortical area (Ct.Ar) and average cortical thickness (Ct.Th) were escalated by R13 after OVX in both WT and BDNF+/− mice; nonetheless, the relative cortical bone area to tissue area (Ct.Ar/Tt.Ar) ratios remained unchanged among the groups (Fig. 3c). Assessments of the serum levels of CTX and osteocalcin indicated osteocalcin was increased in R13-treated OVX mice compared with vehicle-treated sham group, and OVX-treated and R13-

treated OVX mice exhibited higher CTX level compared to the sham group. Nevertheless, OPG were substantially increased upon R13 treatment, leading to significant reduction RANK-L/OPG ratios in both WT and BDNF+/− mice, though the serum BDNF levels remained equivalent among the groups (Fig. 3d). Hence, BDNF haploinsufficiency does not alter femur trabecular bone properties after OVX, but treatments with R13 strongly increase bone density.

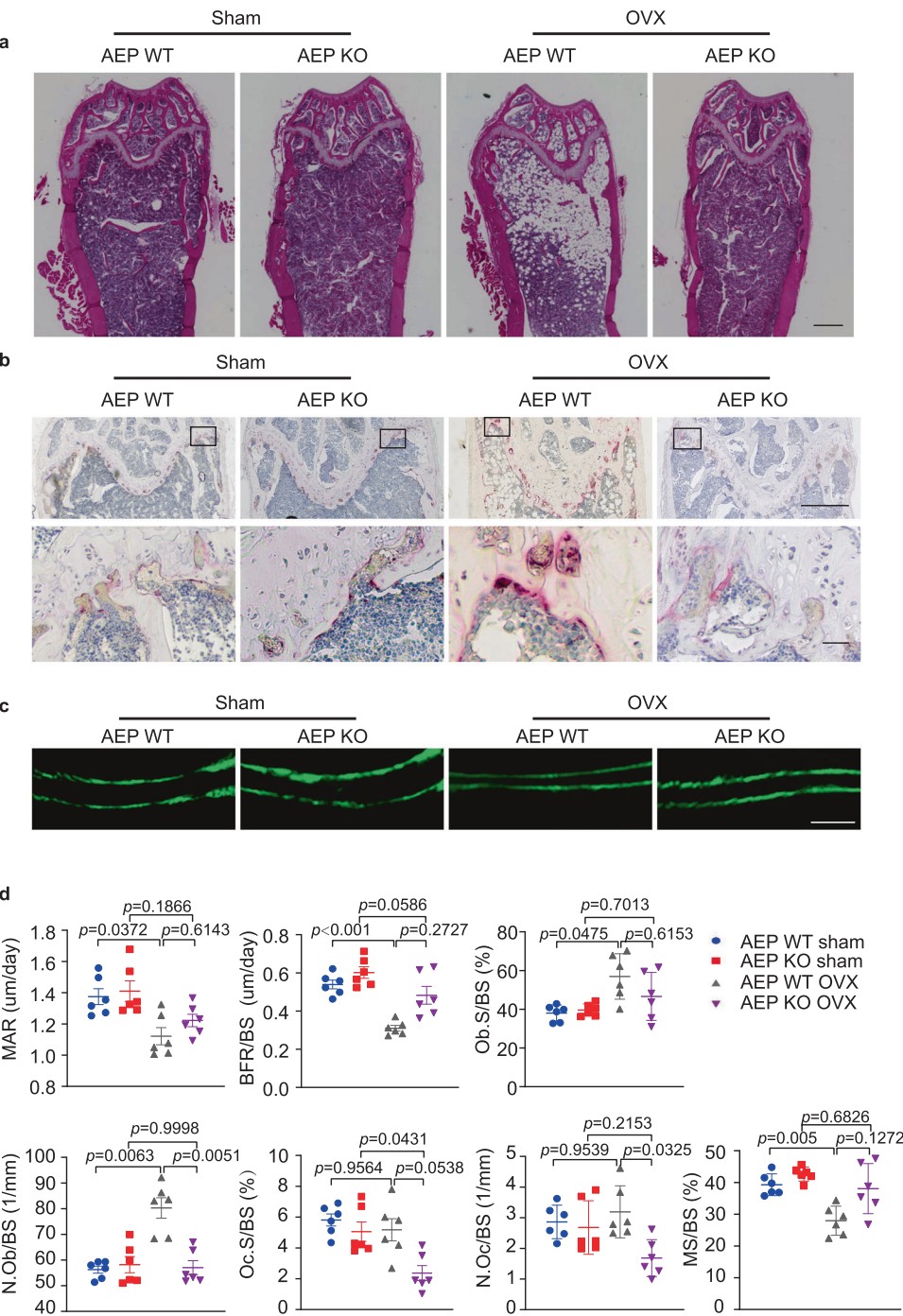

**Fig. 2 | AEP knockout inhibits the bone turnover induced by ovariectomy in female mice. a** Representative images of hematoxylin and eosin (H&E) staining of the distal femur bone in wild-type (AEP WT) and AEP knockout (AEP KO) mice with (OVX) or without (Sham) ovariectomy (*n* = 5 mice per group). (Scale bar, 500 μm). **b** Representative images of tartrate resistant acid phosphatase-stained (TRAP-stained) sections of the distal femur bone in AEP WT sham, AEP KO sham, AEP WT OVX and AEP KO OVX group at low magnification (upper panel) and a selected area (shown as the boxed region in the top row of images) at higher magnification (lower panel) (*n* = 5 mice per group). (Scale bar, 500 μm (upper panel), 20 μm (lower panel)). **c** Mice were injected subcutaneously with calcein at day 10 and day 3 before sacrifice. Trabecular calcein double-fluorescence labeling images (green) of the representative sections in AEP WT sham, AEP KO sham, AEP WT OVX and AEP KO OVX group (*n* = 6 mice per group). (Scale bar, 30 μm). **d** Histomorphometric indices of bone turnover (mineral apposition rate (MAR), bone formation rate per bone surface (BFR/BS), osteoblast surface per bone surface (ObS/BS), number of osteoblasts per bone surface (N.Ob/BS), number of osteoclasts per bone surface (N.Oc/BS), osteoclast surface per bone surface (OcS/BS), and mineralizing surface/ bone surface (MS/BS)) in AEP WT and AEP KO mice with or without ovariectomy. MAR = mineral apposition rate; BFR/BS = bone formation rate; Ob.s/BS = percentage of bone surface covered by osteoblasts; N.Ob/BS = number of osteoblasts per mm bone surface; Oc.S/BS = percentage of bone surface covered by osteoclasts; N.Oc/BS = number of osteoclasts per mm bone surface. MS/BS = mineralizing surface/bone surface (%). Data are shown as mean ± SEM, *n* = 6 mice per group, one-way ANOVA.

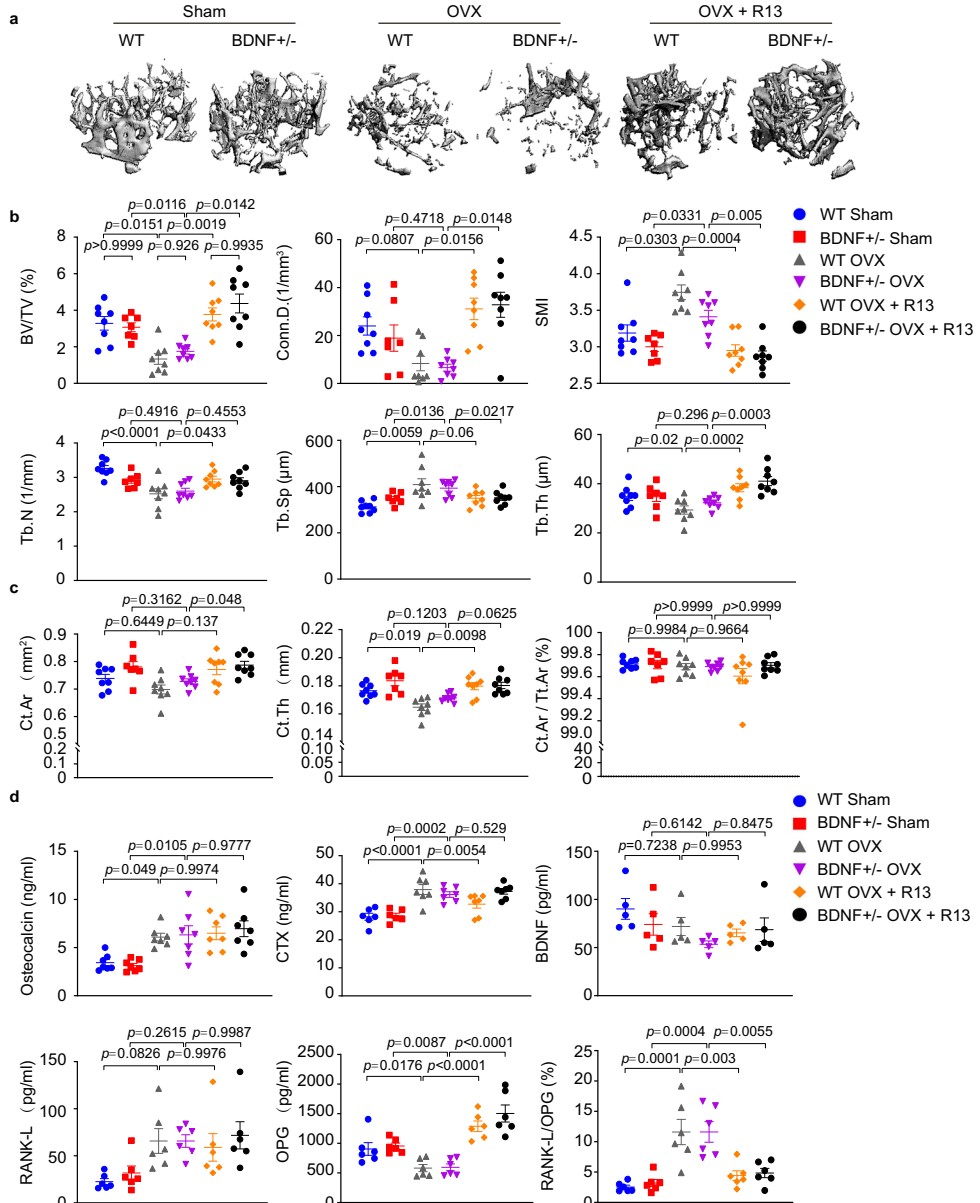

**Fig. 3 | R13 treatment increases serum OPG level and blocks trabecular bone loss induced by ovariectomy in both WT and BDNF+/− female mice.** Femoral bone structures were assessed by in vitro μCT in wild-type and BDNF+/− mice which were with (OVX) or without (sham) ovariectomy at 12 weeks old, and some of which administrated R13 (21.8 mg/kg) or vehicle for 8 weeks (6 days per week) by oral gavage. **a** Representative images of the femoral indices of trabecular bone structure measured by in vitro μCT scan in WT sham, BDNF+/− sham, WT OVX, BDNF+/− OVX, and WT OVX + R13, BDNF+/− OVX + R13 group. **b** μCT scanning measurements of BV/TV, Conn.D., SMI, Tb.N, Tb.Sp, Tb.Th. Data are shown as mean ± SEM, $n = 8$ mice per group ($n = 7$ mice for BDNF+/− sham group), one-way ANOVA. **c** μCT scanning measurements of cortical bone Ct.Ar, Ct.Th and Ct.Ar/Tt.Ar. Data are shown as mean ± SEM, $n = 8$ mice per group ($n = 7$ mice per group for BDNF+/− sham group), one-way ANOVA. **d** Serum levels of osteocalcin, CTX, RANK-L, OPG, RANK-L/OPG ratio, and serum BDNF level. Data are shown as mean ± SEM, $n = 5$ mice per group for BDNF measurement, $n = 6$ mice per group for RANK-L and OPG measurement, $n = 7$ mice per group for osteocalcin and CTX measurement, one-way ANOVA.

To explore whether orally administrated R13 released sufficient 7,8-DHF in the bone marrow to trigger osteoblast differentiation, we conducted in vivo PK (pharmacokinetics) study and found that 7,8-DHF distribution was time-dependently increased in the bone marrow and it reached 31.7 ng/ml (~125 nM), which is much higher than the $EC_{50}$ (~50 nM) for activating TrkB in primary neurons[28]. Its concentration started to increase in the plasma after oral gavage and climaxed around 15–30 min and declined since then, and reached 17.1 ng/ml at 120 min (Supplementary Fig 2a). Immunoblotting of p-TrkB signaling in the bone marrow showed that 7,8-DHF concentrations correlated with neurotrophic signaling activation (Supplementary Fig 2b).

Thus, R13-derived 7,8-DHF activates BDNF/TrkB signaling in the bone marrow after oral administration of R13.

### R13 blocks the changes in bone turnover induced by ovariectomy

To further explore the roles of BDNF signaling in bone resorption and formation after OVX, we conducted H&E staining and analyzed bone morphology and white adipocytes in both WT and BDNF+/− mice after OVX surgery. Cleary, R13 treatment increased trabecular bone tissue and decreased the adipocyte content in the bone after OVX (Fig. 4a). TRAP staining revealed that OVX-induced demonstrable osteoclast

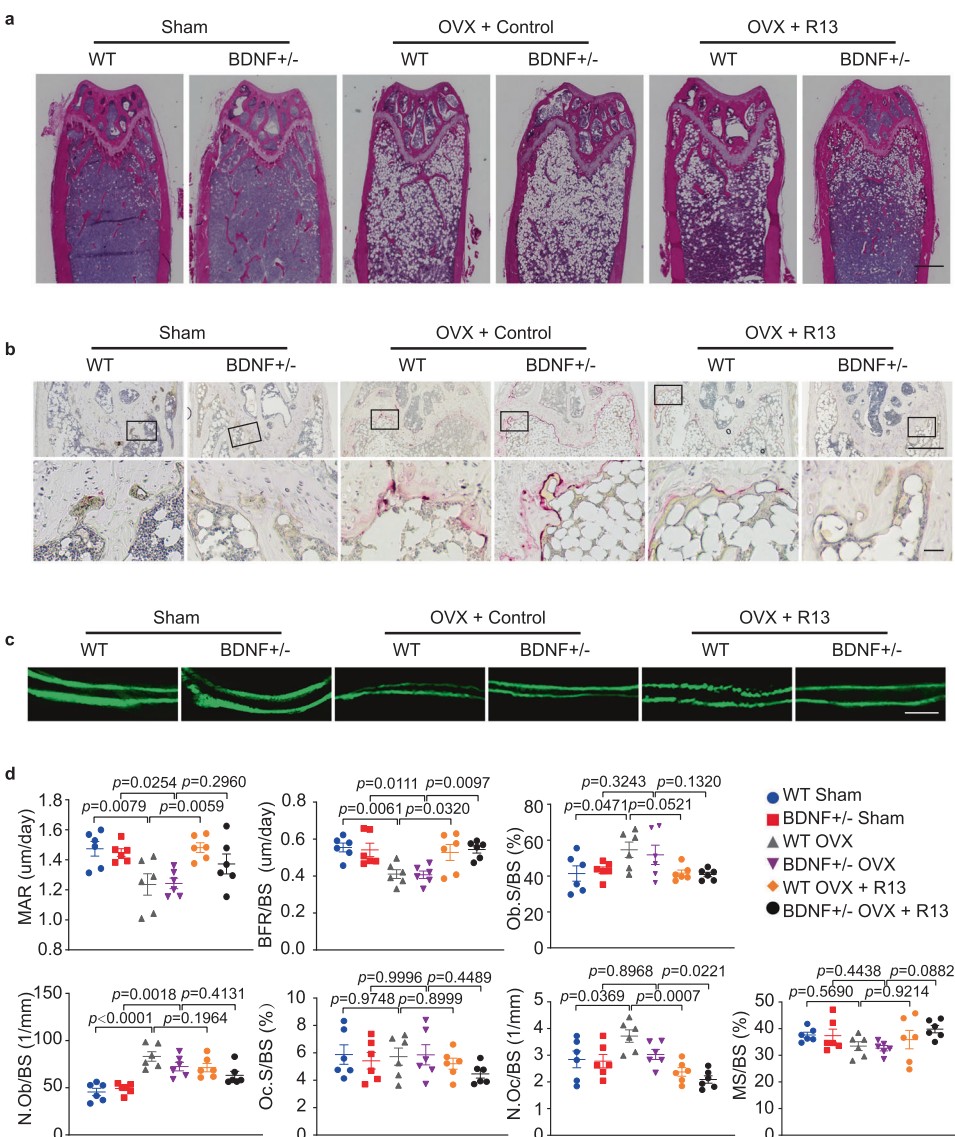

**Fig. 4 | R13 treatment blocks the changes in bone turnover induced by ovariectomy in female mice. a** Representative images of hematoxylin and eosin (H&E) staining of the distal femur bone in WT sham, BDNF+/− sham, WT OVX, BDNF+/− OVX, and WT OVX + R13, BDNF+/− OVX + R13 group (n = 5 mice per group). (Scale bar, 500 μm). **b** Representative images of tartrate resistant acid phosphatase-stained (TRAP-stained) sections of the distal femur bone in WT sham, BDNF+/− sham, WT OVX, BDNF+/− OVX, and WT OVX + R13, BDNF+/− OVX + R13 group shown at low magnification (upper panel) and higher magnification (lower panel) (n = 5 mice per group). (Scale bar, 500 μm (upper panels), 20 μm (lower panels)). **c** Mice were injected subcutaneously with calcein at day 10 and day 3 before sacrifice. Representative images of calcein double-fluorescence labeling images (green) of the trabecular bone in WT sham, BDNF+/− sham, WT OVX, BDNF+/− OVX, and WT OVX + R13, BDNF+/− OVX + R13 group (n = 6 mice per group) (Scale bar, 30 μm). **d** Histomorphometric indices of bone turnover in WT and BDNF+/− mice after OVX with or without R13 treatment. N.Oc/BS and Oc.S/BS are indices of bone resorption. N.Ob/BS, Ob.S/BS, MAR, BFR/BS and MS/BS are indices of bone formation. Data are shown as mean ± SEM, n = 6 mice per group, one-way ANOVA.

cells in both WT and BDNF+/− mice were diminished by R13 treatment (Fig. 4b). Dynamic indices of bone formation showed that vehicle-treated mice exhibited lower MAR and BFR as compared to R13-treated mice after OVX. Representative data of double labeling in trabecular bone are shown in Fig. 4c, indicating that OVX-induced bone loss is attenuated by R13 treatment via an increase in bone formation. These observations were also validated by the single and double-labeled surface and inter-label thickness analysis (Supplementary Fig. 3b). By contrast, no significant differences in ObS/BS and N.Ob/BS, which are static indices of bone formation, were found in vehicle and R13-treated mice. Treatments with R13 inhibited N.Oc/BS in both WT and BDNF+/− mice after OVX. Nonetheless, the percentage of surfaces covered by OCs (Oc.S/BS) and MS/BS ratios remained comparable among the groups regardless of the treatment (Fig. 4d). Hence, these data support that R13 treatment induces bone formation and inhibits bone

resorption after OVX. Remarkably, R13 significantly increased OPG level without affecting RANK-L and it also elevated BV/TV ratio in WT mice without any surgery (Supplementary Fig. 4). Thus, R13 treatment greatly blocks the bone loss induced by OVX and substantially elevates OPG level.

### 7,8-DHF promotes MC3T3-E1 cell differentiation, mineralization, and OPG secretion

7,8-DHF binds to TrkB receptor extracellular region, where BDNF interacts on the TrkB receptors[29], mimicking the biological actions of BDNF in a TrkB-dependent manner[28,30]. To examine the molecular mechanisms of how 7,8-DHF stimulates bone density elevation in rodents, we tested its effect in MC3T3-E1 cells in the presence of OIM (osteogenic induction medium). ALP staining showed that OIM treatment at 14 days evidently enhanced osteoblast cell differentiation,

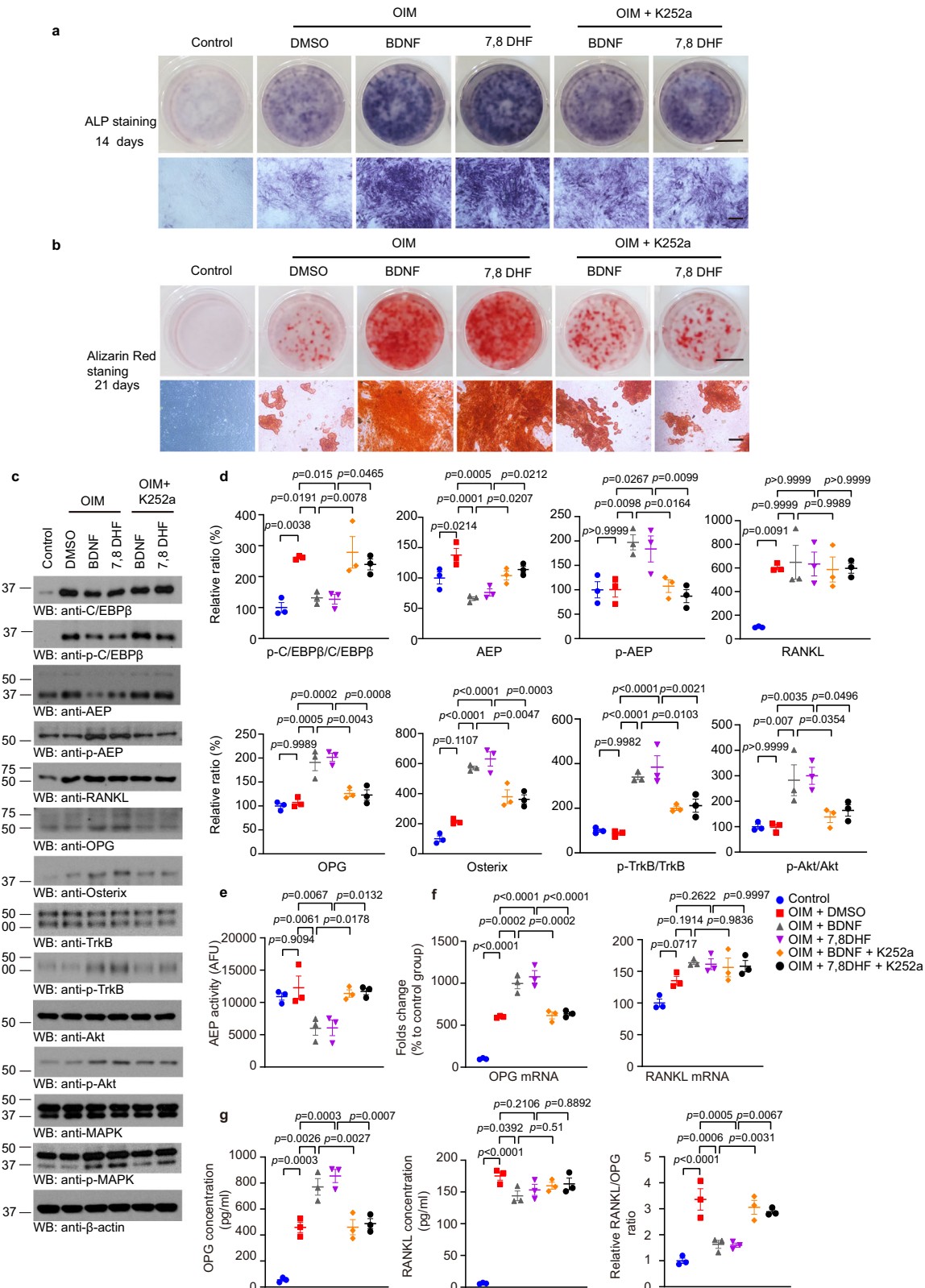

which was further escalated by BDNF or 7,8-DHF, respectively. Alizarin Red staining also validated these observations at 21 days (Fig. 5a, b), supporting the conclusion that BDNF or 7,8-DHF strongly stimulates MC3T3-E1 differentiation and calcium deposition.

Stimulation of the BDNF/TrkB pathway inhibits AEP activation via Akt phosphorylation of T322 residue, sequestering AEP into the lysosomes[24], and decreases AEP expression level via repressing its transcription factor C/EBPβ[26]. Immunoblotting revealed that OIM robustly induced p-C/EBPβ and total C/EBPβ expression in MC3T3-E1 cells, both of which were distinctly repressed by either BDNF or 7,8-DHF. Both of them activated TrkB signaling, leading to p-Akt escalation and its downstream target p-AEP T322 augmentation. Consequently, expression of the downstream effector, AEP, was clearly diminished, which inversely associated with RANK-L and OPG augmentation.

**Fig. 5 | 7,8-DHF promotes MC3T3-E1 cells differentiation, mineralization and OPG secretion. a** Representative images of ALP staining in MC3T3-E1 cells treated with BDNF or 7,8-DHF combined with or without K252a for 14 days (n = 3 independent experiments) (Scale bar, 5 mm (upper panel), 200 μm (lower panel)). **b** Representative images of Alizarin Red S mediated calcium staining in MC3T3-E1 cells treated with BDNF or 7,8-DHF combined with or without K252a for 21 days showed that 7,8-DHF promoted MC3T3 cells mineralization (n = 3 independent experiments) (Scale bar, 5 mm (upper panel), 200 μm (lower panel)). **c** MC3T3 cells were cultured in complete medium or osteogenic induction medium (OIM) with BDNF or 7,8 DHF combined with or without K252a for 4 days. Western blotting results showed 7,8-DHF inhibited C/EBPβ/AEP pathway and increased OPG expression, and K252 inhibited the effect of 7,8-DHF. **d** Relative protein levels of C/EBPβ, p-C/EBPβ, AEP, RANKL, OPG, Osterix, p-TrkB/TrkB, and p-Akt/Akt in MC3T3 cells cultured in the complete medium or OIM with BDNF or 7,8-DHF combined with or without K252a for 4 days. Data are shown as mean ± SEM of three independent experiments, one-way ANOVA; **e** AEP enzymatic activity assay. BDNF and 7,8-DHF inhibit AEP activities, and K252a abolish BDNF and 7,8-DHF's effects. Data are shown as mean ± SEM of three independent experiments, one-way ANOVA. **f** qPCR results show that OPG mRNA expression increases in MC3T3 cells after 7,8-DHF treatment for 4 days. Data are shown as mean ± SEM of three independent experiments, one-way ANOVA. **g** 7,8-DHF increases OPG level and decreases the RANK-L/OPG ratio. Levels of OPG and RANK-L secreted into the medium were measured by ELISA. Data are shown as mean ± SEM of 3 independent experiments, one-way ANOVA.

Osterix, a key early gene in the bone formation cascade, is usually used as a predictive measure of bone formation. As expected, OIM prominently elevated osterix levels as compared with vehicle. Similar findings occurred in the presence of BDNF or 7,8-DHF. Consequently, these effects triggered by BDNF or 7,8-DHF were potently blunted by Trk receptors inhibitor K252a (Fig. 5c, d). In alignment with active AEP repression by BDNF or 7,8-DHF, the enzymatic assay validated that AEP protease activities were greatly blocked (Fig. 5e).

To further interrogate the role of AEP in MC3T3-E1 cell differentiation and mineralization induced by OIM, we transfected the cells with dominant-negative enzymatic-dead AEP C189S mutant and phosphorylation mimetic AEP T322E mutant (inactive), and found that blockade of AEP highly escalated fibronectin, osterix and RUNX2, which were repressed by AEP WT (Supplementary Fig. 5a, d). ALP staining and Alizarin Red S analysis showed that antagonizing AEP strongly promoted osteoblast cell differentiation and bone formation (Supplementary Fig. 5b, c). As expected, AEP C189S and T322E mutants but not WT robustly inhibited OIM-elicited AEP activities (Supplementary Fig. 5e). Quantitative RT-PCR (qRT-PCR) analysis revealed that both BDNF and 7,8-DHF exhibited strong stimulatory effect in promoting OPG mRNA level, which were abolished by K252a. Interestingly, the elevated RANK-L mRNAs by BDNF and 7,8-DHF were not affected by K252a (Fig. 5f). Both OPG and RANK-L protein levels were elevated by OIM in ELISA assays. These elevations were further augmented in the presence of 7,8-DHF or BDNF (Fig. 5g, left two panels), consistent with the findings in Western blotting. Though both RANK-L and OPG concentrations were substantially elevated by BDNF and 7,8-DHF, the ratios of RANK-L/OPG triggered by OIM alone were significantly higher than BDNF and 7,8-DHF. Notably, K252a treatment highly augmented the ratios in BDNF and 7,8-DHF groups (Fig. 5g, right panel). Together, these observations strongly support that 7,8-DHF mimics BDNF and that both strongly escalate OPG expression and decrease RANKL/OPG ratio, accelerating osteoblast formation. Moreover, it also represses the C/EBPβ/AEP pathway, leading to inhibition of osteoclast formation.

### 7,8-DHF increases OPG expression via activating transcription factor CREB

To further interrogate the molecular mechanism of 7,8-DHF in promoting OPG expression, we conducted a time course study in MC3T3-E1 cells in the presence of OIM. As expected, 7,8-DHF swiftly activated p-TrkB and its downstream effectors p-MAPK and p-Akt, supporting that 7,8-DHF indeed mimics BDNF by activating TrkB neurotrophic signaling. Numerous transcription factors including c-Jun and CREB have been shown to be implicated in OPG mRNA transcription[31,32]. Noticeably, p-C/EBPβ, p-c-Jun, and p-CREB signals were time-dependently increased by 7,8-DHF (Fig. 6a, b), suggestive of the activation of these transcription factors. To examine which of them are essential for OPG expression, we knocked down each of them in MC3T3-E1 cells via the specific siRNAs in the presence of OIM and 7,8-DHF. Consistently, OIM manifestly increased OPG and RANK-L, associated with C/EBP and c-Jun augmentation, whereas CREB total level remained constant. Again, 7,8-DHF treatment attenuated C/EBPβ

without interfering CREB or c-Jun levels. Remarkably, knocking down CREB or c-Jun but not C/EBPβ clearly reduced OPG protein levels, and the ratio of RANK-L/OPG was significantly augmented when CREB was depleted (Fig. 6c, d). qRT-PCR demonstrated that 7,8-DHF-stimulated OPG mRNA was selectively suppressed when CREB was knocked down by its siRNA, whereas RNAK-L mRNA levels were similar among the groups, resulting in higher RANK-L/OPG ratio (Fig. 6e). Hence, 7,8-DHF via activating CREB, a well-characterized downstream transcription factor of BDNF/TrkB pathway, stimulates OPG expression levels.

### 7,8-DHF inhibits RANK-L-induced RAW264.7 osteoclastogenesis

RAW264.7 cell is a well-established cellular model for osteoclastic differentiation, which has been widely engaged in bone homeostasis research. Moreover, RANK-L independently induces RAW264.7 cell osteoclastic differentiation, which efficiently generates osteoclasts in vitro[33]. To investigate whether the promotion of bone formation by 7,8-DHF also involves inhibiting osteoclastogenesis, we employed RAW264.7 cells in the presence of RANK-L. Treatment with 30 ng/ml RANK-L at day 4 significantly increased the number of multinucleated osteoclastic cells and this increase was diminished by addition of BDNF or 7,8-DHF, indicating the inhibition of RANK-L promoting osteoclastogenesis (Supplementary Fig. 6a). Immunoblotting analysis revealed that C/EBPβ was greatly reduced by 7,8-DHF or BDNF treatments, and RANK-L-stimulated AEP echoed the upstream C/EBPβ levels (Supplementary Fig. 6b, c). Hence, 7,8-DHF blocks RANK-L-induced RAW264.7 osteoclastogenesis associated with AEP inhibition.

Given that R13 protects the bone by increasing OPG, we employed the anti-RANK-L antibody as a comparator drug. Four weeks after ovariectomy, WT mice were treated with IgG or anti-RANK-L monoclonal antibody consecutively for 4 weeks as previously reported[34]. Remarkably, R13 displayed the similar efficacy in the bone density and various bone indices to anti-RANK-L treatment (Fig. 7a–c). Again, R13 robustly elevated OPG without altering RANK-L, whereas anti-RANK-L substantially depleted RANK-L without changing OPG, resulting in the significant reduction in the ratios of RANK-L/OPG by both treatments (Fig. 7d). These findings support that R13 exhibits the same therapeutic efficacy toward osteoporosis as anti-RANK-L. Given R13 is approved by FDA for the clinical trial for AD indication, we expect that R13 may act as a new therapeutic agent soon for treating osteoporosis via both stimulating bone formation by enhancing osteoblast differentiation and preventing bone resorption via blocking osteoclastogenesis. Because R13 exerts the bone protective effect via the dual mechanisms including OPG upregulation and AEP antagonism, conceivably, it may display even stronger therapeutic efficacy in patients than anti-RANK-L. H&E staining revealed that R13 treatment and anti-RANK-L increased trabecular bone marrow density and decreased the white adipocyte contents in the bone after OVX (Fig. 8a). TRAP staining demonstrated that OVX-induced osteoclast cells were evidently diminished by both treatments (Fig. 8b). Histomorphometric analysis in the trabecular bone in the distal femur demonstrated that R13-treatment significantly increased BFR/BS and decreased N.Oc/BS in WT mice after OVX. However anti-RANKL antibody treatment significantly decreased the

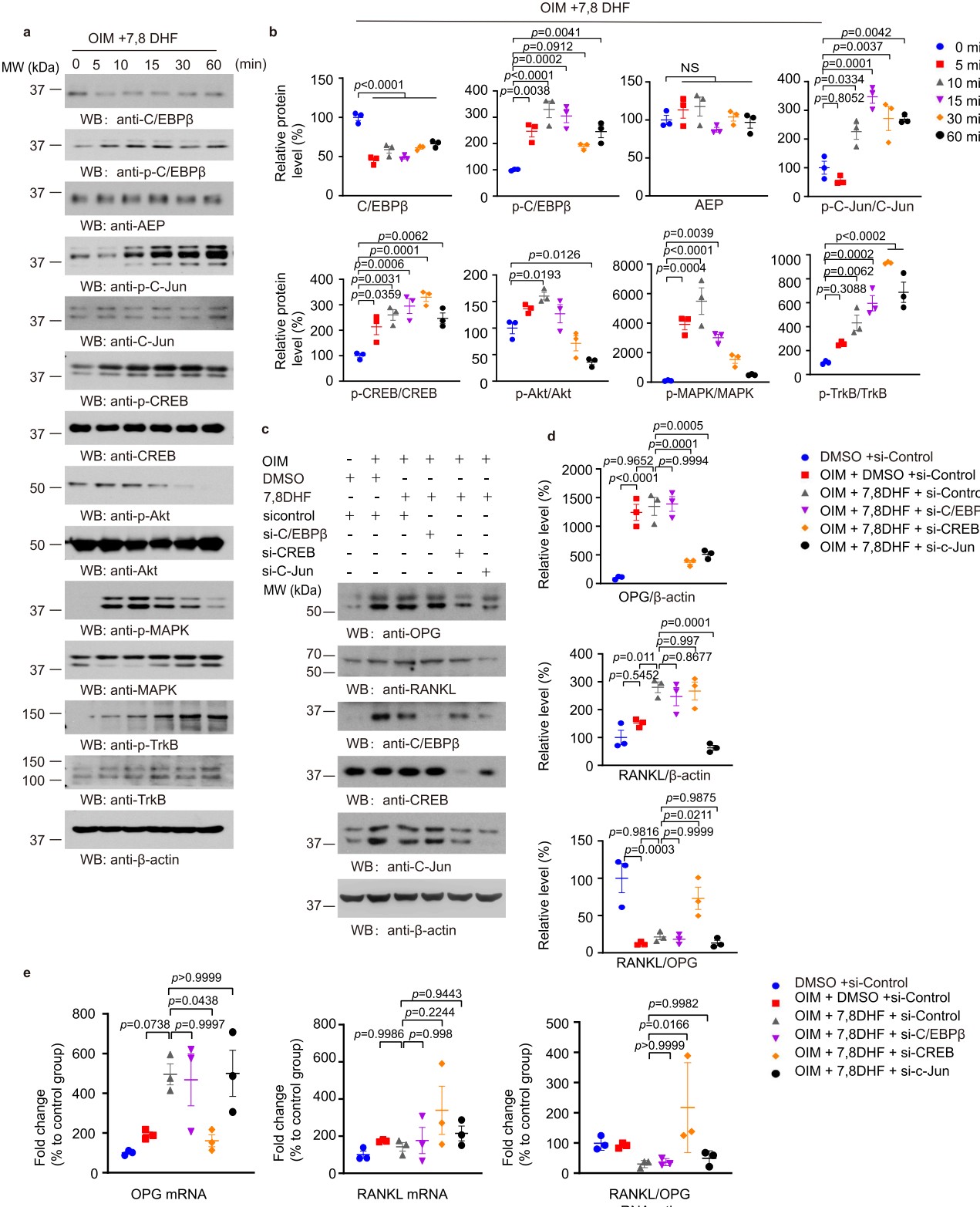

**Fig. 6 | 7,8-DHF positively regulates OPG expression via activating CREB.**
**a** MC3T3 cells cultured in OIM were treated with 7,8-DHF in different time points. Western blotting showed that 7,8-DHF inhibited C/EBPβ, increased Akt (S473), MAPK (p38), C-Jun, CREB phosphorylation. **b** Relative protein level of C/EBPβ, p- C/EBPβ, AEP, phosphorylated C-Jun, CREB, Akt, MAPK, and TrkB in MC3T3 cells treated with 7,8-DHF in different time points. Data are shown as mean ± SEM of three

independent experiments, one-way ANOVA. **c** Western blotting showed that knockdown of CREB blunted 7,8-DHF-induced OPG expression. **d** Relative protein levels of RANKL, OPG, and RANKL/OPG ratio. Data are shown as mean ± SEM of three independent experiments, one-way ANOVA. **e** qPCR results showed that knockdown of CREB inhibited OPG mRNA expression induced by 7,8-DHF. Data are shown as mean ± SEM of three independent experiments, one-way ANOVA.

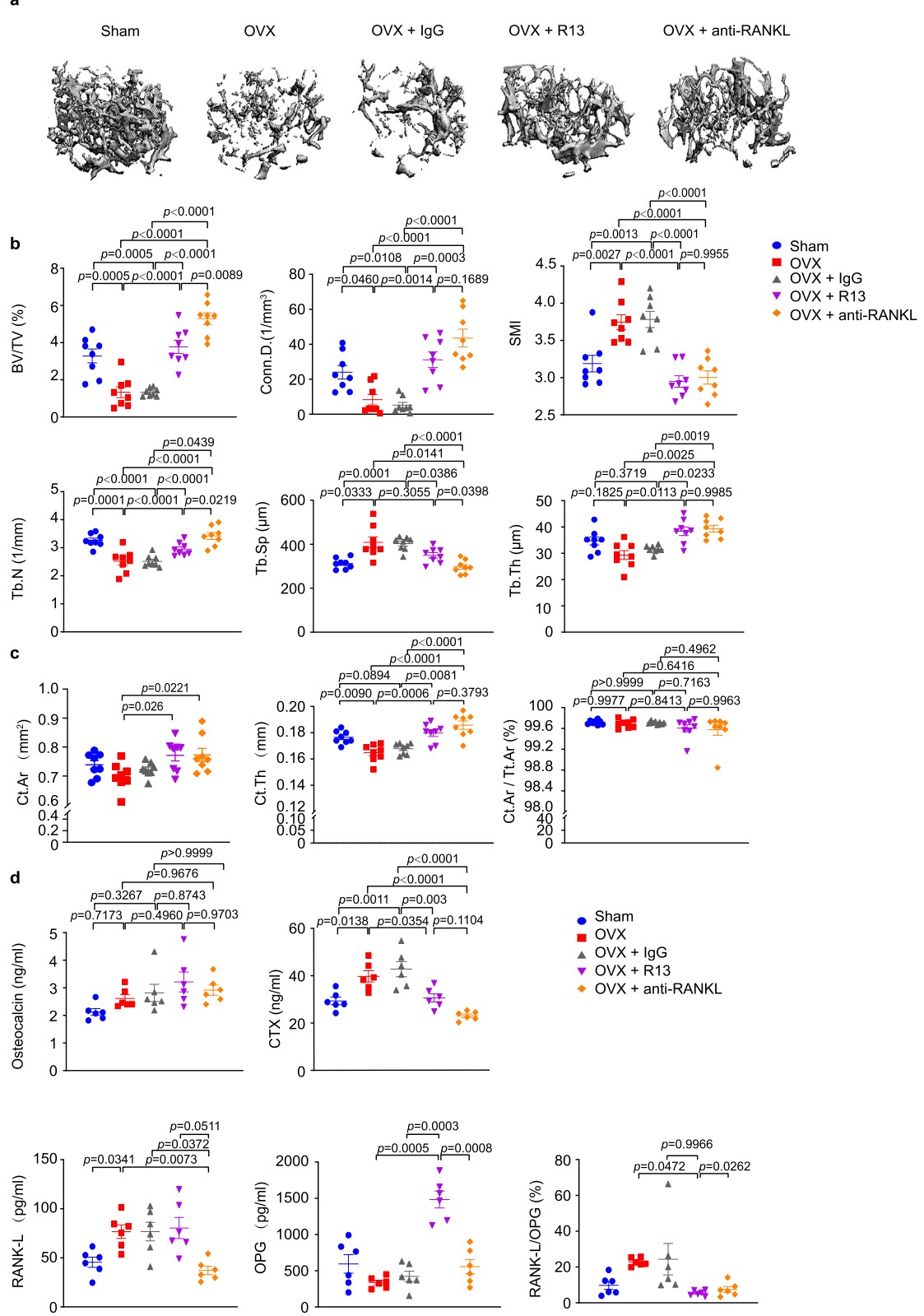

**Fig. 7 | R13 and anti-Rank-L antibody treatments display the similar effect on blocking trabecular bone loss induced by ovariectomy in WT female mice.** Femoral bone structures were assessed by in vitro μCT in wild-type mice which were obtained ovariectomy at 12 weeks old, and some of which were treated with R13 (21.8 mg/kg, by oral gavage) or anti-RANK-L antibody. **a** Representive images of the femoral indices of trabecular bone structure measured by in vitro μCT scan.

**b** μCT scanning measurements of BV/TV, Conn.D., SMI, Tb.N, Tb.Sp, Tb.Th. Data are shown as mean ± SEM, $n = 8$ mice per group, one-way ANOVA. **c** μCT scanning measurements of cortical bone Ct.Ar, Ct.Th and Ct.Ar/Tt.Ar. Data are shown as mean ± SEM, $n = 8$ mice per group, one-way ANOVA. **d** Serum levels of osteocalcin, CTX, RANK-L, OPG and RANK-L/OPG ratio. Data are shown as mean ± SEM, $n = 6$ mice per group, one-way ANOVA.

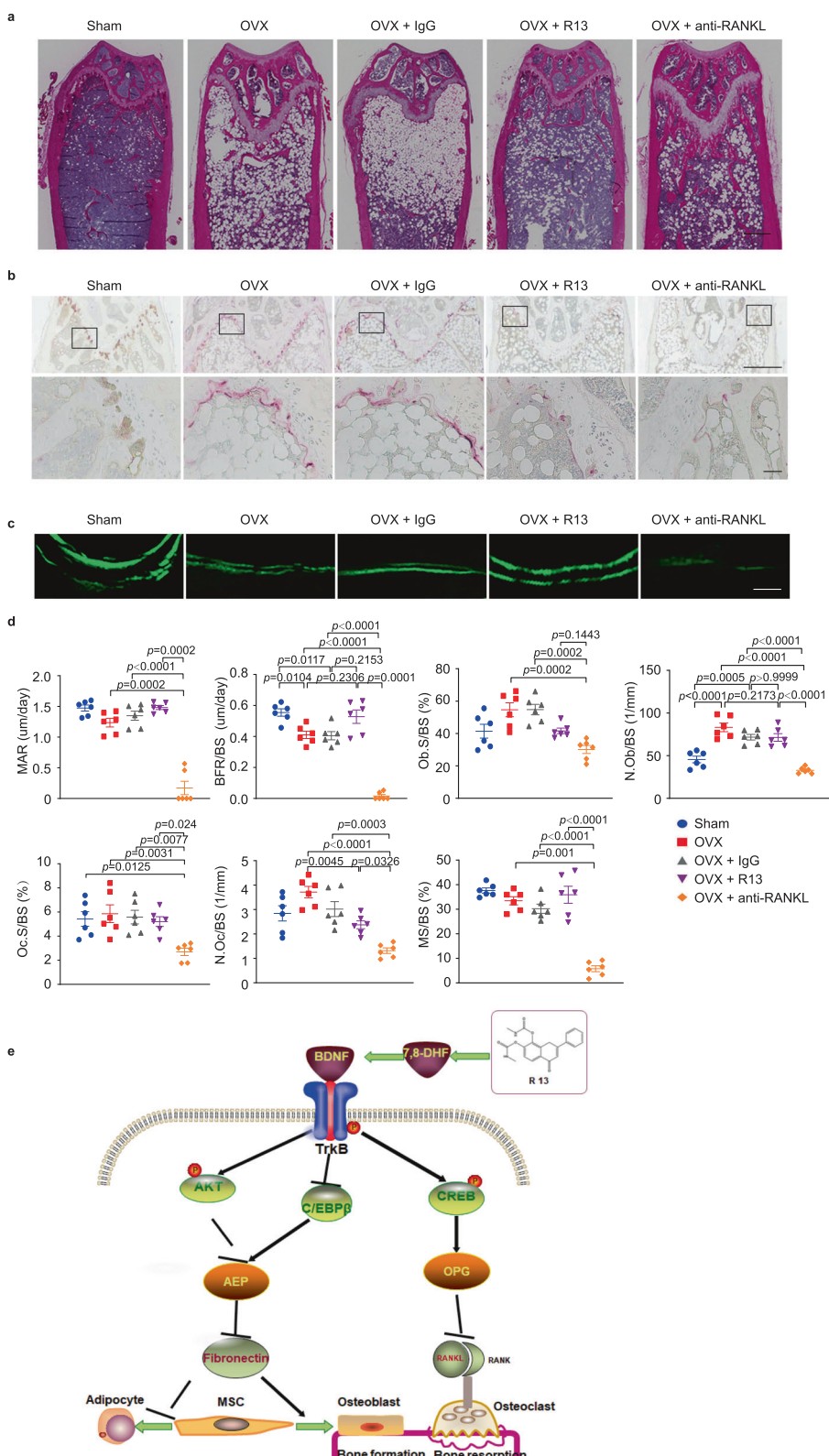

**Fig. 8 | R13 and anti-RANK-L antibody treatments block the changes in bone turnover induced by ovariectomy in female mice. a** Representative images of hematoxylin and eosin (H&E) staining of the distal femur bone in WT mice with sham, OVX, OVX + IgG, OVX + R13, and OVX + anti-RANK-L antibody group (*n* = 5 mice per group). (Scale bar, 500 μm). **b** Representative images of tartrate resistant acid phosphatase-stained (TRAP-stained) sections of the distal femur bone were shown at low magnification (upper panel) and higher magnification (lower panel) (*n* = 5 mice per group). (Scale bar, 500 μm (upper panels), 20 μm (lower panels)). **d** Representive calcein double-fluorescence labeling images of the trabecular bone (*n* = 6 mice per group) (Scale bar, 30 μm). **d** Histomorphometric indices of distal femur. N.Oc/BS and Oc.S/BS are indices of bone resorption. N.Ob/BS, Ob.S/BS, MAR, BFR/BS, and MS/BS are indices of bone formation. Data are shown as mean ± SEM, *n* = 6 mice per group, one-way ANOVA. **e** The schematic diagram of R13 treatment on osteoporosis via elevating OPG and inhibiting AEP via activating BDNF/TrkB signaling.

MAR, BFR/BS, ObS/BS, and N.Ob/BS which are the bone formation parameters, as well as the bone resorption parameters such as OcS/BS and N.Oc/BS. These results suggested that R13 exhibits comparable effect as anti-RANK-L in reducing OVX-induced bone loss by inhibiting osteoclast (Fig. 8d). Representative data of double labeling of the trabecular bone in WT mice with OVX after R13 or anti-RANKL antibody treatment are provided in Fig. 8c.

## Discussion

BDNF stimulates mRNA expression of the osteoblast differentiation marker, osteocalcin, and promotes the differentiation of MC3T3-E1 cells, augmenting new bone formation and maturation[8]. Both BDNF and its TrkB receptor are demonstrable in various stages of the bone formation process in human fracture gap tissues and upregulated in human osteoblasts[7]. However, we found that BDNF +/− mice fail to exhibit any significant difference in bone loss from WT littermates upon OVX, indicating that endogenous BDNF/TrkB pathway might be dispensable in preventing bone loss triggered by OVX. Nevertheless, TrkB receptor agonist R13 treatment substantially elevates OPG and reduces RANK-L/OPG ratios in both WT and BDNF +/− mice after OVX, leading to prominent bone density augmentation (Figs. 3 and 4). Previous study shows that central BDNF deletion produces a marked skeletal phenotype characterized by increased femur length, elevated whole bone mineral density, and bone mineral content. Moreover, the skeletal changes are developmentally regulated and appear concurrently with the metabolic phenotype, suggesting that the metabolic and skeletal actions of BDNF are linked[35]. However, we did not find any significant bone alteration in BDNF+/− mice as compared to WT littermates (Fig. 3). Presumably, complete BDNF knockout in the CNS may disrupt the endocrine hormones that mediate the metabolism and bone homeostasis.

Accumulative evidence supports that BDNF/TrkB signaling is suppressed in depression. In addition to the previously proposed neurotransmitter deficiency hypothesis in depression field, the "neurotrophine" hypothesis has been advocated in the past decade. Indeed, patients with osteoporosis display depressive symptoms. There is a growing body of evidence that depression impacts the risk for fracture in older adults[36]. Mounting evidence shows that BDNF is age-dependently decreased in human brains. BDNF and its TrkB receptor are demonstrated in various stages of the bone formation process, and they are upregulated in human osteoblasts and implicated in fracture healing[7]. BDNF strongly elevates mRNA expression of osteoblast differentiation marker, osteocalcin, in MC3T3-E1 cells. Moreover, BDNF stimulates the differentiation of MC3T3-E1 cells and promotes new bone formation and maturation[8]. Conceivably, depression-associated BDNF reduction may decrease osteoblast differentiation and reduce new bone formation, facilitating osteoporosis.

The proinflammatory cytokines (e.g., TNF-α and IL-6) are upregulated in osteoporotic bone marrow microenvironment[37]. These cytokines activate the transcription factor C/EBPβ, which feeds back and acts as transcription factor for these cytokines as well[38]. Most recently, we show that BDNF and C/EBPβ mutually regulate each other negatively. For instance, BDNF deficiency increases production of inflammatory cytokines and activates the JAK2/STAT3 pathway, resulting in the upregulation of transcription factor C/EBPβ[26]. In turn, C/EBPβ acts a repressor that binds to BDNF exon IV promoter and blocks BDNF mRNA transcription[39]. Treatments with BDNF and 7,8-DHF thus pronouncedly diminish C/EBPβ expression, leading to AEP reduction, which is inversely correlated with RANK-L and OPG escalation (Figs. 5 and 6; Supplementary Fig. 4). Interestingly, we observed OPG elevation in AEP KO mice as compared to WT mice under sham condition (Fig. 1d), suggesting that AEP somehow physiologically represses OPG expression. Previously, we reported that AEP cuts α-Synuclein N103 and Tau N368, which bind to the TrkB receptor intracellular domain and inhibit the neurotrophic activities[40,41].

Conceivably, AEP antagonizes BDNF/TrkB neurotrophic signalings, leading to OPG suppression. Depletion of AEP consequently alleviates the inhibition and escalates OPG expression. OPG plays a suppressive role in cytokine-induced osteoclastogenesis[42]. Moreover, both CREB and c-fos transcription factors mediate OPG and RANKL mRNA expression[31]. Notably, CREB, a crucial downstream transcription factor of BDNF/TrkB pathway, plays a pivotal role in mediating 7,8-DHF-stimulated OPG escalation, though all of transcription factors including C/EBPβ, c-Jun and CREB are phosphorylated upon 7,8-DHF treatment, accompanied by p-TrkB and its downstream effectors escalation (Fig. 6a, b). Thus, 7,8-DHF-triggered p-CREB is indispensable for augmenting OPG expression and osteoblast differentiation.

AEP is a secreted cysteine protease involved in diverse biological processes. The proteolytic activity of AEP is important for its effects on hBMSC differentiation and bone formation, and AEP inhibits osteoblast cell differentiation through degradation of fibronectin[12]. AEP expression is elevated in hBMSCs from osteoporotic patients and, at single-cell resolution, and AEP overexpression in adipocyte differentiation is inversely correlated with local trabecular bone volume. Previous study shows that C-terminus of AEP blocks osteoclast (OCL) formation and bone resorption. Notably, AEP significantly inhibits OCL-like multinucleated cell formation induced by 1,25-dihydroxyvitamin D(3) and parathyroid hormone-related protein in bone marrow cultures, and bone resorption in a dose-dependent manner[10]. X-ray crystallography study shows that the C-terminus of AEP acts as an inhibitory domain for antagonizing AEP protease activity[43]. Fragmentation of the C-terminus of AEP from the rest of the protease yields an active enzymatic form, which exerts the osteoblast differentiation inhibiting activity (Fig. 5 and Fig. S5). In alignment with these findings, the inhibitory C-terminus fragment of AEP blocks the osteoclast formation, in addition to blockade of the protease activity of active form of AEP. Hence, blockade of AEP activation provides an innovative pharmacological interference strategy for treating osteoporosis. Noticeably, CTX is increased in both WT and AEP KO mice after OVX, while the ratios of RANK-L/OPG were significantly different between WT and AEP KO mice (Fig. 1D). It is well established that several cytokines in addition to RANK-L/OPG ratio contribute to the bone loss induced by OVX. For example, we recently reported that TNF and IL-17 play an essential role in OVX-induced bone loss[44]. We thus interpret our findings as being consistent with the fact that TNF, IL-17 (but also IL-6) are implicated in OVX-induced bone loss.

Recently, we have reported that C/EBPβ upregulates AEP expression during aging[25]. BDNF or 7,8-DHF robustly represses C/EBPβ expression induced by OIM (Osteogenic induction medium) or RANK-L, resulting in AEP reduction and its protease activity repression. Consequently, 7,8-DHF strongly blocked RANK-L induced RAW264.7 osteoclastogenesis (Supplementary Fig. 6). Remarkably, we have recently reported that 7,8-DHF decreases follicle stimulatory hormone (FSH) production, resulting in increased serum estradiol in female mice treated with HFD[45]. Previously, it has been reported that FSH triggers bone loss and anti-FSH increased bone density without altering estrogen concentrations[46,47]. It is possible that FSH might somehow activate AEP and trigger bone loss. Imaginably, 7,8-DHF represses FSH production, resulting in AEP inhibition and bone density increase. Clearly, the data presented above, combined with AEP KO mice display diminished bone loss upon OVX and reduced RANK-L/OPG ratios, strongly support the conclusion that R13 may ameliorate OVX-induced bone loss via antagonizing AEP and elevating OPG (Fig. 8e).

## Methods
### Animals
Female C57BL6/J wild-type mice and BDNF+/− mice were obtained from Jackson Laboratory (MMRRC stock#000664 and 002267), then held and underwent breeding at Emory School of Medicine. The AEP knockout mice on a mixed C57BL/6 and 129/Ola background were

generated as reported[48]. All in vivo experiments were carried out in female mice. All mice were kept under specific pathogen-free conditions in an environmentally controlled clean room with the humidity ranged 40% to 60% and housed at 22 °C on a 12-h/12-h light/dark cycle. Food and water were provided ad lib. The experiments were conducted according to the NIH animal care guidelines and Emory School of Medicine guidelines. The protocol was reviewed and approved by the Institutional Animal Care and Use Committee (IACUC) at Emory University. All procedures performed in studies involving animals were in accordance with the ethical standards of the Emory Institutional Animal Care and Use Committee. WT, BDNF+/− mice, AEP WT, and AEP knockout mice were bilaterally ovariectomized or sham operated at 12 weeks of age. Four days after ovariectomy, the WT and BDNF+/− mice received vehicle or R13 dissolved in 5% DMSO/0.5% methylcellulose at dose of 21.8 mg/kg/d, six days per week, for 8 weeks by gavage. In another group, 4 weeks after ovariectomy, WT mice were treated with IgG or anti-RANK-L monoclonal antibody (10 mg/kg, twice per week) by single subcutaneous injection for 4 weeks.

## Cell culture
Murine MC3T3-E1 (subclone 4) cells and RAW 264.7 cells were obtained from American Type Culture Collection (ATCC, Manassas, VA, USA, catalog#: CRL-2593 and catalog#: TIB-71). The MC3T3-E1 cells were cultured in alpha-MEM with 10% FBS and 0.1% penicillin/streptomycin, but without ascorbic acid. The RAW 264.7 cells were cultured in DMEM supplemented with 10% FBS and 0.1% penicillin–streptomycin. The cells were maintained at 37 °C in a humidified atmosphere of 95% air and 5% $CO_2$.

## Antibody and reagents
Antibody to C/EBPβ (HT-7) (catalog#: sc-7962, 1:1000 for western blotting), RANKL (catalog#: sc-377079, 1:1000 for western blotting), OPG (catalog#: sc-390518, 1:1000 for western blotting), osterix (catalog#: sc-393325, 1:1000 for western blotting) and RUNX2 (catalog#: sc-101145, 1:1000 for western blotting) was from Santa Cruz; antibodies to Legumain (D6S4H) (catalog#: 93627, 1:2000 for western blotting), p-C/EBPβ(catalog#: 3084s, 1:1000 for western blotting), Akt (catalog#: 4691s, 1:2000 for western blotting), p-Akt(S473) (catalog#: 9271s, 1:1000 for western blotting), MAPK (catalog#: 9102s, 1:1000 for western blotting), p-MAPK (catalog#: 9106s, 1:1000 for western blotting), p-c-Jun (Ser73, 1:1000 for western blotting) (catalog#: 3270T), c-Jun (catalog#: 9165T, 1:1000 for western blotting), CREB (catalog#: 9197T, 1:1000 for western blotting) and p-CREB (catalog#: 9198T, 1:1000 for western blotting) were purchased from Cell Signaling Technology; Antibody to TrkB (catalog#: MAB397, 1:1000 for western blotting) was from R&D System; antibody to β-actin (catalog#: A5316, 1:3000 for western blotting) and Fibronectin (catalog#: F3648, 1:1000 for western blotting) were from Sigma-Aldrich; Antibodies to p-AEP (T322) (1:1000 for western blotting) and p-TrkB (Tyr816) (1:1000 for western blotting) were developed in the Ye lab; Anti-mouse IgG-HRP (catalog#: 70765, 1:3000 for western blotting) and anti-rabbit IgG-HRP (catalog#: 70745, 1:3000 for western blotting) were from Cell Signaling Technology). Anti-RANKL monoclonal antibody (catalog#: 510012) and anti-IgG antibody (catalog#: 401412) for mice treatment is obtained from Ichorbio. Alpha-MEM (catalog#: A1049001) were obtained from Gibco.Lipo3000 transfection reagent (catalog #: L3000008) were obtained from Invitrogen. K252a (catalog#: ab120419) were obtained from Abcam. TRACP&ALP double-staining kit (catalog#: MK300) was from TakaRa Bio.Alkaline Phosphatase Assay kit (catalog#: ab83369) were from Abcam. The AEP substrate Z-Ala-Ala-Asn-AMC (catalog#: 4033201) was from Bachem, and EZ-Link Sulfo-NHS-LC-Biotinylation Kit (catalog #: 21435) was obtained from Thermo Fisher. Serum Osteocalcin elisa kit (catalog#: NBP2-68151) were from Novus biologicals, CTX elisa kit (catalog#: AC-06F1) was from Immunodiagnostic

systems, RANKL (catalog#: ab269553) and OPG (catalog#: ab203365) was from Abcam. All chemicals not mentioned above were purchased from Sigma-Aldrich.

## Osteogenic differentiation
MC3T3-E1 cells were seeded into plates in complete medium and cultured for 24 days until the cells reached 70% confluence. To initiate the differentiation, the cells were incubated in osteogenic induction medium (OIM) containing α-MEM, 10% FBS, dexamethasone (10-7M), β-glycerophosphate (10 mM) and ascorbic acid (50 µg/ml). The differentiation medium was replaced every 3 days, with DMSO, BDNF (50 ng/ml), or 7,8 DHF (0.5 µM) added into the medium with or without the pretreatment of K252a (100 nM). The MC3T3-E1 cells were transfected with AEP C189S, AEP WT, AEP T322E plasmid, C/EBP β siRNA (sc-29862, Santa Cruz Biotechnology, USA), CREB siRNA (sc-35111, Santa Cruz Biotechnology, USA), C-Jun siRNA (sc-29224, Santa Cruz Biotechnology, USA) or control plasmid or control-siRNA (sc-44237, Santa Cruz Biotechnology, USA) by Lipo3000 transfection reagent according to the instructions.

## Osteoclast differentiation
RAW264.7 cells were seeded in 24 wells plates and cultured for 24 h in DMEM with 10% FBS and 0.1 penicillin/streptomycin. The medium was changed to α-MEM with 5% FBS, 0.1% penicillin/streptomycin. The receptor activator of NF-κB ligand (RANKL, 30 ng/ml) was added to induce osteoclast differentiation. The medium was replaced every 3 days, accompanied with DMSO, BDNF (50 ng/ml) or 7,8 DHF (0.5 µM) added into the medium.

## ALP staining
MC3T3-E1 cells were plated in 24-well plates, cultured in complete medium or OIM, and treated with BNDF (50 ng/ml) or 7,8 DHF (0.5 µM) with or without the pretreatment of K252a (100 nM) for 14 days. The Cells were washed in PBS twice, and fixed for 10 min with fixing buffer at room temperature, stained the ALP staining with the TRACP&ALP double-staining kit.

## Alizarin Red S staining
MC3T3-E1 cells were plated in 24-well plates, cultured in complete medium or OIM, and treated with BNDF (50 ng/ml) or 7,8 DHF (0.5 µM) with or without the pretreatment of K252a (100 nM), and then were washed in distilled water twice and fixed in 70% ice-cold ethanol. Then the cells were stained with 2% Alizarin Red S solution to detect calcification.

## TRAP staining
RAW 264.7 cells were cultured in α-MEM with or without RANKL, in the presence or absence of BNDF (50 ng/ml) or 7,8 DHF (0.5 µM) for 5 days. The cells were washed in PBS twice, fixed in fixing solution for 10 min at room temperature, and then stained the TRAP activity with the TRACP&ALP double-staining kit according to the supplied protocols.

## ALP activity assay
MC3T3-E1 cells were incubated in osteogenic induction medium (OIM), followed with the treatment of different doses of 7,8-DHF (0, 5, 10, 50, 100, 500, 1000 ng/ml) for 14 days. The cells were washed with cold PBS and harvested in assay buffer. The ALP enzyme activities were analyzed by Alkaline Phosphatase Assay kit according to the manufacture's instruction.

## In vivo PK of 7,8-DHF
Two months old female mice were treated with oral administration of R13 (21.8 mg/kg), and then were sacrificed to collect the serum and bone marrow at different time points (0, 15, 30, 60, 120 min) after

oral gavage of R13 with 3 mice/ group. Concentrations of 7,8 DHF in the plasma and bone marrow samples were determined by LC-MS/MS.

## Western blotting

MC3T3-E1 and RAW 264.7 cells were washed with ice-cold PBS and lysed in (50 mM Tris, pH 7.4, 40 mM NaCl, 1 mM EDTA, 0.5% Triton X-100, 1.5 mM Na3VO4, 50 mM NaF, 10 mM sodium pyrophosphate, 10 mM sodium β-glycerophosphate, supplemented with protease inhibitors cocktail) at 4 °C for 0.5 h, and centrifuged for 25 min at 15,000 rpm. The supernatant was boiled in SDS loading buffer. After SDS-PAGE, the samples were transferred to a nitrocellulose membrane. The membrane was blocked with TBS containing 5% nonfat milk and 0.1% Tween 20 (TBST) at room temperature for 2 h, followed by the incubation with primary antibody at 4 °C overnight, and with the secondary antibody at room temperature for 2 h. After washing with TBST, the membrane was developed using the enhanced chemiluminescent detection system.

## AEP activity assay

Cell lysates (10 µg) were incubated in 200 µl assay buffer (20 mM citric acid, 60 mM Na2HPO4, 1 mM EDTA, 0.1% CHAPS, and 1 mM DTT, pH 6.0) containing 20 µM δ-secretase substrate Z-Ala-Ala-Asn-AMC (Bachem). AMC released by substrate cleavage was quantified by measuring at 460 nm in a fluorescence plate reader at 37 °C for 2 h in kinetic mode.

## Quantitative real-time PCR analysis

Total RNA was isolated by TRIzol (Life Technologies). Reverse transcription was performed with SuperScript III reverse transcriptase (Life Technologies). Gene-specific primers and probes were designed and bought from Taqman (Life Technologies). All real-time PCR reactions were performed using the ABI 7500-Fast Real-Time PCR System and the Taqman Universal Master Mix Kit (Life Technologies). The relative quantification of gene expression was calculated using the ΔΔCt method. We use predesigned real-time PCR primers from Applied Biosystems for the analysis of Opg (Tnfrsf11b; Mm0043545_m1), Rankl (Tnfsf11; Mm00441908_m1), AEP (Lgmn; Mm01325350_m1), GAPDH (Gapdh; Mm99999915_g1).

## µCT measurements

µCT scan and analysis were performed in femurs ex vivo using a µCT-40 scanner, as previously reported[49,50]. Voxel sizes were 12 µm$^3$ for the in vitro measurements of femurs. For the femoral trabecular region, we analyzed 140 slices, beginning 50 slices below the distal growth plate. X-ray tube potential was 70 kVp, and integration time was 200 ms. Representative samples were reconstructed in 3D to generate visual representations of trabecular structure.

## Quantitative bone histomorphometry

The measurements, indices, and units for histomorphometric analysis were recommended by the Nomenclature Committee of the American Society of Bone and Mineral Research[51]. Mice were injected with calcein (25 µg/g) subcutaneously at day 10 and day 3 before sacrifice. Bone histomorphometric analysis was performed at the University of Alabama at Birmingham Center for Metabolic Bone Disease-Histomorphometry and Molecular Analysis Core Laboratory. The Goldner's trichrome-stained plastic-embedded sections of calcein-double labeled femora of the mice were analyzed by an operator blinded as to the nature of the samples.

## Biochemical markers of bone turnover

Serum Osteocalcin, CTX, RANKL, and OPG were measured by specific Elisa assays.

## Statistics and reproducibility

Information on biological replicates (n) is indicated in the figure legends. All statistical analyses were performed by GraphPad Prism 9 software. A one-way analysis of variance was used to determine whether there was a significant difference in experiments with more than two groups and two-tailed upaired Student's $t$ test were used to determine whether there was a significant difference in experiments between two groups.

## Reporting summary

Further information on research design is available in the Nature Research Reporting Summary linked to this article.

## Data availability

The data supporting the findings from this study are available within the manuscript and its supplementary information. Source data are provided with this paper.

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

## Acknowledgements

This work was supported by grants from NIH grant (RF1, AG051538) to K.Y. The authors are thankful for Dr. Arthur W. English at Cell Biology Department at Emory University for critical proofreading the manuscript.

## Author contributions

Y.K. conceived the project, designed the experiments, analyzed the data, and wrote the manuscript. X.J and L.J.M. designed and performed most of the experiments. L.X. prepared the animal breeding. A.J. performed the in vitro bone CT analysis. Z.Z. and P.R. contributed to write the manuscript.

## Competing interests

The authors declare no competing interests.
