## [Peer Review File · Nature Communications]

REVIEWER COMMENTS

Reviewer #1 (Remarks to the Author):

BNDF triggers trkB receptor activation in human osteoclasts leading to inactivation of AEP resulting in bone proliferation and formation. The authors describe a new drug R13 that mimic the action of BNDF. R13 by delivering 7,8-DHF triggers trkB receptor dimerization resulting in inhibition of C/EBP β transcription factor resulting in inactivation of AEP and bone formation. Using several deficient mice: BNDF $^{-/-}$, AEP KO $^{-/-}$ and an in vivo model of bone loss (ovariectomy), they clearly demonstrate the therapeutic effect of this new permeable drug R13 in osteoporosis.

AEP is a cysteine protease, which has many substrates and blocks osteoclast formation through fibronectin degradation. Activation of AEP (inactive form 57kDa) requires the sequential removal of the C (inactive form 47kDa) and N (active form 46 kDa) terminal domains at acidic PH by auto cleavage at asparagine 323 and 25 respectively. Further trimming can occur yielding to a fully active form running at a size of 36kDa .

In the introduction, paragraph dedicated to AEP, can the authors explain a little bit more the role of AEP C-terminal fragment (17 kDa size) which, inhibits osteoclast differentiation. This AEP C-terminal fragment should run at 11kDa and not 17kDa (Li DN et al, JBC, 2003). In none of the WB, the C-terminal fragment of AEP is shown.

AKT phosphorylates AEP at residue T322 that enables AEP autocatalytic cleavage? Is this correct and how does this work? In Figure 5C, M3C3-E4 cells express the active form of AEP (36kDa) at the steady state. Stimulation with OIM slightly increases its expression, which is down regulated upon BDNF or 7,8-DHF treatment (less 36kDa form detected). According to previous publication, upon BDNF treatment AKT phosphorylates AEP at residue T322 that triggers its inactivation and translocation to lysosome (In Figure 2J, Wang ZH et al, JCI insight 2018, phosphorylated AEP runs at 57kDa or 56 kDa). If this is the same mechanism here describe in this paper (in addition to the repression of the transcription factor C/EBP β), why is the fully active form of AEP detected here (36kDa)? WB of phosphorylated form of AEP should be shown.

Furthermore, in Supp Fig 3, use of the AEP mutant T322E which can not be phosphorylates should be include in the experiment as well as control WT AEP.

In the WB, I presume that AEP C189S can be cleaved by endogenous AEP?

Overall, detailed mechanisms of AEP inactivation (phosphorylation or not) should be addressed to make the paper stronger.

Minor comments

Figure 5 MC3T3-E4 (osteoblast lineage cell) cultured in osteogenic medium for 4 days show upregulation of RANKL, CEBP β , pCEBP β , OPG and down regulation of AEP. Similar AEP expression is detected in cells with or without osteogenic medium. In Supp Fig3, using the same cell line, AEP expression differs a lot with or without osteogenic medium. Is there an explanation for this?

Reviewer #2 (Remarks to the Author):

This manuscript demonstrates that R13, a prodrug of 7,8-DHF may ameliorate OVX-induced bone loss via antagonizing AEP and elevating OPG. The study has several major limitations that are indicated below.

- 1. Bones have not been studied with rigor. Include BMC data in Fig 1 and Fig 3. What part of the femur was taken; proximal or distal? What happens to other cancellous bones such as the lumbar vertebra? What happens to cortical bones (femur mid-diaphysis)?**
- 2. Lack of bone strength data does not allow us to determine if the microarchitectural changes that are shown here translate to gain in strength by the treatment.**
- 3. Fig 1; how to explain increased CTX-1 in AEP KO OVX sham compared with AEP KO while RAKL/OPG ratio was not significant between the two groups?**
- 4. Fig 2C and 4C: Representative images give me an impression that cortical bone sections have been shown while trabecular bones have been studied (mentioned in the Results section). Where are MS/BS data? Please provide single- and double labelled surface and**

inter-label thickness data as supplementary.

5. Fig 2D: MAR (the mean speed at which individual osteoid are mineralized) and BFR/BS (amount of new bone formed per unit of bone surface over unit time) are decreased in AEP WT OVX compared with AEP WT Sham and yet osteoblast surface and number are higher in AEP WT OVX over AEP WT Sham. Remarkably, osteoclast surface and number are remarkably unchanged between AEP WT Sham and AEP WT OVX that do not explain increased CTX and RANKL/OPG ratio in the latter over the former. Doesn't high turnover bone loss that characterizes OVX typically display increased osteoclast number and surface which is responded by increased osteoblast number? Taken together, there is a lack of internal consistency in the data.

6. Data are poorly organized; WT sham and BDNF sham data are missing in Fig 4.

7. Fig 5: 0.5 microM 7,8-DHF was used to study the differentiation of MC3T3-E1 cells. What is the EC50 of this compound? What is the bone marrow level of 7,8-DHF upon single oral dosing of R13 at 21.8mg/kg? Does that reach around E50 of 7,8 DHF? It is important to determine a preclinical PK-PD relationship concerning the use of R13 in osteoporosis treatment.

8. Given that R13 protects bone by increasing OPG, the anti-RANKL antibody should have been used as a comparator drug.

Reviewer #3 (Remarks to the Author):

The authors showed that R13, a prodrug for 7,8-DHF, inhibited AEP and promotes bone formation. The experiments were conducted well. The data of this paper are interesting for the readers of the journal. However the following minor concerns should be addressed.

Minor concerns:

1) BDNF(+/-) mice did not show any significant difference in bone loss from WT mice upon OVX. BDNF (+/-) mice are reported to be obesity. Did you measure body weight of the groups?

2) Figure 5: BDNF or 7,8-DHF significantly stimulated MC3T3-E1 differentiation and calcium deposition. Are these effects blocked by TrkB antagonist?

3) Patients with osteoporosis have depressive symptoms. Decreased BDNF-TrkB signaling plays a role in depression which could be involved in bone diseases. Please discuss the role of BDNF-TrkB signaling in comorbidity of depression and bone disease.

EMORY
UNIVERSITY
SCHOOL OF
MEDICINE

Department of Pathology and Laboratory Medicine

Keqiang Ye, Ph.D.

Professor

Room 141, Whitehead Building

615 Michael Street

Atlanta, GA 30322

Telephone: (404) 712-2814 / Fax: (404) 712-2979

E-mail: kye@emory.edu

December 22nd, 2021

Dr Maria-Teresa Piccoli
Senior Editor, Nature Communications
Nature Research

Dear Dr. Piccoli,

Thank you very much for monitoring our manuscript entitled “**R13, a TrkB Agonist Prodrug, Inhibits Asparagine Endopeptidase (AEP) and Increases Osteoprotegerin (OPG), Preventing Bone Loss**” (manuscript # 2020-02510). We appreciate you for providing us an opportunity to address the concerns raised by the referees. We have carefully digested the comments from the 3 referees and are grateful for the constructive comments from the reviewers. We have addressed the referees' concerns with additional experimentation. The revised parts are highlighted in RED. The point-by-point response to the 3 referees' comments is listed as follows:

Reviewer #1:

BDNF triggers trkB receptor activation in human osteoclasts leading to inactivation of AEP resulting in bone proliferation and formation. The authors describe a new drug R13 that mimic the action of BDNF. R13 by delivering 7,8-DHF triggers trkB receptor dimerization resulting in inhibition of C/EBP β transcription factor resulting in inactivation of AEP and bone formation. Using several deficient mice: BDNF^{-/-}, AEP KO^{-/-} and an in vivo model of bone loss (ovariectomy), they clearly demonstrate the therapeutic effect of this new permeable drug R13 in osteoporosis. AEP is a cysteine protease, which has many substrates and blocks osteoclast formation through fibronectin degradation. Activation of AEP (inactive form 57kDa) requires the sequential removal of the C (inactive form 47KDa) and N (active form 46 kDa) terminal domains at acidic PH by auto cleavage at asparagine 323 and 25 respectively. Further trimming can occur yielding to a fully active form running at a size of 36kDa.

1. In the introduction, paragraph dedicated to AEP, can the authors explain a little bit more the role of AEP C-terminal fragment (17 kDa size) which, inhibits osteoclast differentiation. This AEP C-terminal fragment should run at 11kDa and not 17kDa (Li DN et al, JBC, 2003). In none of the WB, the C-terminal fragment of AEP is shown.

A: AEP is processed into enzymatically active 36 kDa mature form, as well as a 11 kDa C-terminal inhibitory fragment¹. Strikingly, the C-terminal truncate inhibits osteoclast differentiation through binding to an uncharacterized receptor^{2,3}. We don't show the C-terminus of AEP in immunoblotting, because our antibodies can't recognize it.

Previous study shows that C-terminus of AEP blocks osteoclast (OCL) formation and bone resorption. Notably, AEP significantly inhibits OCL-like multinucleated cell formation induced by 1,25-dihydroxyvitamin D (3) and parathyroid hormone-related protein (PTHrP) in bone marrow cultures, and bone resorption in a dose-dependent manner. X-ray crystallography study shows that the C-terminus of AEP acts as an inhibitory domain for antagonizing AEP protease activity. Fragmentation of the C-terminus of AEP from the rest of the protease yields an active enzymatic form, which exerts the osteoblast differentiation inhibitory activity (Fig 5 and Fig S5). In alignment with these findings, the inhibitory C-terminus fragment of AEP blocks the osteoclast formation, in addition to blockade of the protease activity of active form of AEP. Hence, blockade of AEP activation provides an innovative pharmacological interference strategy for treating osteoporosis (Discussion, page 18).

2. AKT phosphorylates AEP at residue T322 that enables AEP autocatalytic cleavage? Is this correct and how does this work? In Figure 5C, M3C3-E4 cells express the active form of AEP (36kDa) at the steady state. Stimulation with OIM slightly increases its expression, which is down regulated upon BDNF or 7,8-DHF treatment (less 36kDa form detected). According to previous publication, upon BDNF treatment AKT phosphorylates AEP at residue T322 that triggers its inactivation and translocation to lysosome (In Figure 2J, Wang ZH et al, JCI insight 2018, phosphorylated AEP runs at 57kDa or 56 kDa). If this is the same mechanism here describe in this paper (in addition to the repression of the transcription factor C/EBPb), why is the fully active form of AEP detected here (36kDa)? WB of phosphorylated form of AEP should be shown.

A: In our previous paper, we demonstrate that Akt phosphorylates AEP on T322 and inhibits its autocleavage and blocks AEP activation. AEP is a cysteine protease that shreds the substrates after asparagine residue and it has no stringent peptide sequence in the substrates. However, it preferentially cuts the substrates at NK sites with neutral or positive charge at P-1 or -2 positions. Because the autocleavage site locates at N323, which is right next to the phosphorylation residue by Akt, the negative charge from phosphate on p-T322 prevents AEP cysteine protease autocleavage. MC3T3 cells possess high basal active AEP levels and subchronic BDNF or 7,8-DHF treatment (4 days) robustly represses active AEP formation. As requested, we present that p-AEP T322 signals associated with the upstream p-Akt/Akt levels in the revised Fig 5C.

3. Furthermore, in Supp Fig 3, use of the AEP mutant T322E which can not be phosphorylates should be include in the experiment as well as control WT AEP.

In the WB, I presume that AEP C189S can be cleaved by endogenous AEP?

Overall, detailed mechanisms of AEP inactivation (phosphorylation or not) should be addressed to make the paper stronger.

A: As suggested, we included AEP T322E mutant (T322E mimics Akt-phosphorylated AEP) and AEP WT in revised Fig S5 and found that inactivation of AEP by transfected with AEP C189S and AEP T322E can increase fibronectin, osterix and RUNX2 expression, promoting the osteoblast differentiation.

Yes, as the referee indicates that AEP C189S mutant can be shredded by endogenous active AEP on N323 residue to yield 36 kDa **enzymatic-dead** C189S “truncate form”.

We have published several articles regarding AEP post-translational modification to regulate its activation. For instance, we show that SRPK2, a cell cycle-mediated kinase, phosphorylates AEP on S226 residue and triggers its cytoplasmic translocation from the lysosomes and proteolytic activation to yield 36 kDa active form, getting access to its substrates⁴. Moreover, we also reported that Akt phosphorylated AEP on T322 residue and sequestered its lysosomal residency, blocking its activation⁵. BDNF or 7,8-DHF treatment activates TrkB-mediated PI3K/Akt signaling, leading to p-AEP T322 and inactivation of AEP. Hence, though both C189S and T322E AEP mutants can be cut by endogenous active AEP, they display no enzymatic activities. On the other hand, BDNF/TrkB pathway inhibits C/EBP β , a crucial AEP transcription factor, resulting in AEP expression suppression. Hence, R13 antagonizes AEP through both post-translational phosphorylation and transcriptional repression via activating BDNF/TrkB pathway.

Minor comments

Figure 5 MC3T3-E4 (osteoblast lineage cell) cultured in osteogenic medium for 4 days show upregulation of RANKL, CEBP β , pCEBP β , OPG and down regulation of AEP. Similar AEP expression is detected in cells with or without osteogenic medium. In Supp Fig3, using the same cell line, AEP expression differs a lot with or without osteogenic medium. Is there an explanation for this?

A: In Fig 5C and revised Supplementary Fig 5A, OIM elevates active AEP levels in MC3T3-E4 cells as compared to control. Since inactive AEP C189S is overexpressed, highly yielding abundant enzymatic-dead “truncate form” AEP. In order to show active fragment, we present short time exposure data. In Fig 5C, since both BDNF and 7,8-DHF repressed active AEP fragmentation, in order to present active AEP fragment, we presented long time exposure results, so that the active AEP form appears comparable under control and OIM in these two Figs.

Reviewer #2

This manuscript demonstrates that R13, a prodrug of 7,8-DHF may ameliorate OVX-induced bone loss via antagonizing AEP and elevating OPG. The study has several major limitations that are indicated below.

1. Bones have not been studied with rigor. Include BMC data in Fig 1 and Fig 3. What part of the femur was taken; proximal or distal? What happens to other cancellous bones such as the lumbar vertebra? What happens to cortical bones (femur mid-diaphysis)?

A: The laboratory of Dr. Pacifici has 30 years of experience with the phenotypic characterization of mouse skeleton. Attesting to our expertise are many publications on the effects of ovariectomy (ovx) and PTH on the mouse skeleton including recent papers in Nature Communications⁶ and JCI⁷. For the last 20 years, we used μ CT as main tool to assess indices of trabecular volume and structure, as this method is vastly superior to DXA. Contrary to DXA, CT scanning provides a true 3-dimensional volumetric assessment and indices of trabecular structure measured by CT correlate closely with bone strength measured by biomechanical methods. We have not included BMC in the manuscript, because in our opinion this index does not provide any information in addition to that generated by CT. However, if the Reviewer feels strongly about this, we would re-analyze the bones by DXA and calculate BMC.

As stated in the methods, μ CT measurements were obtained in the distal femur. Vertebrae were not measured as in our experience the spine responds to ovx similarly to the femur. The cortical bone data were shown in Fig 1C and Fig 3C. In Fig 1C, we found AEP knockout can alleviate the cortical bone loss induced by ovariectomy. In Fig 3C, it showed that R13 treatment can partially increase the cortical bone density.

2. Lack of bone strength data does not allow us to determine if the microarchitectural changes that are shown here translate to gain in strength by the treatment.

A: As stated above, direct measurements of bone strength by 3-point bending or 4-point bending are not typically included in studies on the effects of ovx, because μ CT indices of bone volume and structure are excellent predictors of bone strength. In addition, the variability of biomechanical tests is notoriously very large. 10-15 mice per group are typically required for biomechanical studies, a factor that renders these studies expensive, time consuming and hard to justify in terms of animal welfare.

3. Fig 1; how to explain increased CTX-1 in AEP KO OVX sham compared with AEP KO while RANKL/OPG ratio was not significant between the two groups?

A: It is well established that several cytokines in addition to RANK-L/OPG contribute to the bone loss induced by ovx. For example, we recently reported in JCI that TNF and IL-17 play an essential role in ovx induced bone loss⁸. We thus interpret our findings as being consistent with the fact that TNF, IL-17 (but also IL-6) are implicated in ovx-induced bone loss. This is now stated in the revised manuscript (Discussion, page 17–19).

4. Fig 2C and 4C: Representative images give me an impression that cortical bone sections have been shown while trabecular bones have been studied (mentioned in the Results section). Where are MS/BS data? Please provide single- and double labelled surface and inter-label thickness data as supplementary.

A: Yes, trabecular bones were analyzed. The data are shown in Fig 2C and 4C. As requested, we include MS/BS data in the revised Fig 2D, Fig 4D. Moreover, we also provide single and

double-labeled surface and inter-label thickness data in Fig S3.

5. Fig 2D: MAR (the mean speed at which individual osteoid are mineralized) and BFR/BS (amount of new bone formed per unit of bone surface over unit time) are decreased in AEP WT OVX compared with AEP WT Sham and yet osteoblast surface and number are higher in AEP WT OVX over AEP WT Sham. Remarkably, osteoclast surface and number are remarkably unchanged between AEP WT Sham and AEP WT OVX that do not explain increased CTX and RANKL/OPG ratio in the latter over the former. Doesn't high turnover bone loss that characterizes OVX typically display increased osteoclast number and surface, which is responded by increased osteoblast number? Taken together, there is a lack of internal consistency in the data.

A: We respectfully disagree with the comments. In Fig 2D, we quantitatively compared the differences between AEP KO and WT after OVX and reported the dynamic and static indices of bone formation BFR/BS, MAR and N. Ob/BS. We found that both **BFR/BS and MAR indices were significantly decreased in WT mice** albeit N. Ob/BS index was increased, whereas **these effects were abolished in AEP KO mice**, suggesting that bone formation is substantially decreased in WT mice but not in AEP KO mice after OVX. The finding of increased N. Ob/BS in the face of decreased MAR and BFR/BS may suggest that OVX caused an increase in osteoblast number. However, the activity of osteoblasts was decreased by OVX.

On the other hand, for the bone resorption, **N. Oc/BS remained the same in WT mice**, however, **it was significantly reduced in AEP KO mice** after OVX. Though both CTX and RANK-L/OPG ratios were augmented in both strains after OVX, WT mice displayed significantly higher RANK-L/OPG ratio than AEP KO mice. These effects resulted in a net effect that WT exhibited significantly higher N. Oc/BS than AEP KO mice upon OVX. Therefore, though OVX elicited higher N. Ob/BS in WT than AEP KO mice, this effect is not counteracted by both MAR and BFR/BS prominent reduction in WT mice, leading to noticeable bone loss in WT versus AEP KO mice.

6. Data are poorly organized; WT sham and BDNF sham data are missing in Fig 4.

A: As suggested, we have included WT sham and BDNF sham data in revised Fig 4.

7. Fig 5: 0.5 microM 7,8-DHF was used to study the differentiation of MC3T3-E1 cells. What is the EC₅₀ of this compound? What is the bone marrow level of 7,8-DHF upon single oral dosing of R13 at 21.8 mg/kg? Does that reach around EC₅₀ of 7,8-DHF? It is important to determine a preclinical PK/PD relationship concerning the use of R13 in osteoporosis treatment.

A: As requested, we have conducted the PK/PD relationship study using 21.8 mg/kg dose in 3 months old female WT mice after oral administration of R13, and collected the serum and bone marrow at 0, 15, 30, 60 and 120 min after oral gavage of R13 with 3 mice/group. LC/MS/MS analysis reveals that [7,8-DHF] at different time points are: 62.903 ng/ml (15 min), 64.681 ng/ml (30 min), 52.081 ng/ml (60 min), 17.093 ng/ml (120 min) in serum and 31.663 ng/ml (120 min; ~125 nM) in bone marrow,

respectively. This dose fits with our previous finding that ~ 50 nM EC₅₀ for 7,8-DHF to activate TrkB receptors in primary neurons (K_d 15.4 nM)^{9,10}. In addition, we have also monitored p-TrkB/TrkB/p-MAPK/MAPK/p-Akt/Akt signaling in bone marrow samples and found that R13 time-dependently activated TrkB neurotrophic pathway, fitting with in vitro observations (Fig S2). Thus, the PK/PD study strongly supports that oral R13 releases 7,8-DHF that penetrates into the bone marrow and activates TrkB signaling. (page 8-9)

8. Given that R13 protects bone by increasing OPG, the anti-RANKL antibody should have been used as a comparator drug.

A: As requested, we performed anti-RANK-L treatment on WT mice. Four weeks after ovariectomy, WT mice were treated with IgG or anti-RANK-L monoclonal antibody consecutively for 4 weeks as previously reported¹¹. Remarkably, R13 displayed similar efficacy in the bone density and various bone indices as compared to anti-RANK-L treatment (Figure 7A-C). Again, R13 robustly elevated OPG without altering RANK-L, whereas anti-RANK-L substantially depleted RANK-L without changing OPG, resulting in the significant reduction in the ratios of RANK-L/OPG by both treatments (Figure 7D). Bone turnover was analyzed in Figure 8. These findings support that R13 exhibits the similar therapeutic efficacy toward osteoporosis as anti-RANK-L. Given R13 is approved by FDA for clinical trial for AD indication, we expect that R13 may act as a new therapeutic agent in the near future for treating osteoporosis via both stimulating bone formation by enhancing osteoblast differentiation and preventing bone resorption via blocking osteoclastogenesis. Because R13 exerts the bone protective effect via the dual mechanisms including OPG upregulation and AEP antagonism, conceivably, it may display even stronger therapeutic efficacy in patients than anti-RANK-L. (page 14)

Reviewer #3

The authors showed that R13, a prodrug for 7,8-DHF, inhibited AEP and promotes bone formation. The experiments were conducted well. The data of this paper are interesting for the readers of the journal. However the following minor concerns should be addressed.

1) BDNF (+/-) mice did not show any significant difference in bone loss from WT mice upon OVX. BDNF (+/-) mice are reported to be obesity. Did you measure body weight of the groups?

A: BDNF +/- mice indeed displayed obesity as compared to the WT littermates. As suggested, we measured the body weight as shown in the revised Supplementary Fig 1C.

2) Figure 5: BDNF or 7,8-DHF significantly stimulated MC3T3-E1 differentiation and calcium deposition. Are these effects blocked by TrkB antagonist?

A: As suggested, we included Trks antagonist K252a, and found that this inhibitor strongly blocked MC3T3-E1 differentiation elicited by BDNF or 7,8-DHF. The new data are included in revised Fig 5.

3) Patients with osteoporosis have depressive symptoms. Decreased BDNF-TrkB signaling plays a

role in depression, which could be involved in bone diseases. Please discuss the role of BDNF-TrkB signaling in comorbidity of depression and bone disease.

A: Accumulative evidence supports that BDNF/TrkB signaling is suppressed in depression. In addition to the previously proposed neurotransmitter deficiency hypothesis in depression field, the “neurotrophine” hypothesis has been advocated in the past decades. Indeed, patients with osteoporosis display depressive symptoms. There is a growing body of evidence that depression impacts the risk for fracture in older adults¹². Mounting evidence shows that BDNF is age-dependently decreased in human brains. BDNF and its TrkB receptor are demonstrated in various stages of the bone formation process, and they are upregulated in human osteoblasts and implicated in fracture healing¹³. BDNF strongly elevates mRNA expression of osteoblast differentiation marker, osteocalcin, in MC3T3-E1 cells. Moreover, BDNF stimulates the differentiation of MC3T3-E1 cells and promotes new bone formation and maturation¹⁴. Conceivably, depression-associated BDNF reduction may decrease osteoblast differentiation and reduce new bone formation, facilitating osteoporosis. (Discussion page 16)

Once again, thank you very much for monitoring our manuscript.

Best Regards,

Keqiang Ye, Ph.D.

1. Dall, E. & Brandstetter, H. Structure and function of legumain in health and disease. *Biochimie* **122**, 126-150 (2016).
2. Choi, S.J., *et al.* Identification of human asparaginyl endopeptidase (legumain) as an inhibitor of osteoclast formation and bone resorption. *J Biol Chem* **274**, 27747-27753 (1999).
3. Choi, S.J., Kurihara, N., Oba, Y. & Roodman, G.D. Osteoclast inhibitory peptide 2 inhibits osteoclast formation via its C-terminal fragment. *J Bone Miner Res* **16**, 1804-1811 (2001).
4. Wang, Z.H., *et al.* Delta-Secretase Phosphorylation by SRPK2 Enhances Its Enzymatic Activity, Provoking Pathogenesis in Alzheimer's Disease. *Mol Cell* **67**, 812-825 e815 (2017).
5. Wang, Z.H., *et al.* BDNF inhibits neurodegenerative disease-associated asparaginyl endopeptidase activity via phosphorylation by AKT. *JCI Insight* **3**(2018).
6. Yu, M., *et al.* PTH induces bone loss via microbial-dependent expansion of intestinal TNF(+) T cells and Th17 cells. *Nat Commun* **11**, 468 (2020).
7. Li, J.Y., *et al.* Parathyroid hormone-dependent bone formation requires butyrate production by intestinal microbiota. *J Clin Invest* **130**, 1767-1781 (2020).

8. Zaiss, M.M., Jones, R.M., Schett, G. & Pacifici, R. The gut-bone axis: how bacterial metabolites bridge the distance. *J Clin Invest* **129**, 3018-3028 (2019).
9. Jang, S.W., *et al.* N-acetylserotonin activates TrkB receptor in a circadian rhythm. *Proc Natl Acad Sci U S A* **107**, 3876-3881 (2010).
10. Liu, X., *et al.* Biochemical and biophysical investigation of the brain-derived neurotrophic factor mimetic 7,8-dihydroxyflavone in the binding and activation of the TrkB receptor. *J Biol Chem* **289**, 27571-27584 (2014).
11. Tokuyama, N., *et al.* Individual and combining effects of anti-RANKL monoclonal antibody and teriparatide in ovariectomized mice. *Bone Rep* **2**, 1-7 (2015).
12. Mezuk, B., Eaton, W.W. & Golden, S.H. Depression and osteoporosis: epidemiology and potential mediating pathways. *Osteoporos Int* **19**, 1-12 (2008).
13. Kilian, O., *et al.* BDNF and its TrkB receptor in human fracture healing. *Ann Anat* **196**, 286-295 (2014).
14. Ida-Yonemochi, H., Yamada, Y., Yoshikawa, H. & Seo, K. Locally Produced BDNF Promotes Sclerotic Change in Alveolar Bone after Nerve Injury. *PLoS One* **12**, e0169201 (2017).

REVIEWER COMMENTS

Reviewer #1 (Remarks to the Author):

A: In our previous paper, we demonstrate that Akt phosphorylates AEP on T322 and inhibits its autocleavage and blocks AEP activation. AEP is a cysteine protease that shreds the substrates after asparagine residue and it has no stringent peptide sequence in the substrates. However, it preferentially cuts the substrates at NK sites with neutral or positive charge at P-1 or -2 positions. Because the autocleavage site locates at N323, which is right next to the phosphorylation residue by Akt, the negative charge from phosphate on p-T322 prevents AEP cysteine protease autocleavage. MC3T3 cells possess high basal active AEP levels and subchronic BDNF or 7,8-DHF treatment (4 days) robustly represses active AEP formation. As requested, we present that p-AEP T322 signals associated with the upstream p-Akt/Akt levels in the revised Fig 5C.

But in Fig 5C, we do not see more AEP inactive form in cells treated with BDNF or 7,8-DHF.

We have published several articles regarding AEP post-translational modification to regulate its activation. For instance, we show that SRPK2, a cell cycle-mediated kinase, phosphorylates AEP on S226 residue and triggers its cytoplasmic translocation from the lysosomes and proteolytic activation to yield 36 kDa active form, getting access to its substrates⁴. Moreover, we also reported that Akt phosphorylated AEP on T322 residue and sequestered its lysosomal residency, blocking its activation⁵. BDNF or 7,8-DHF treatment activates TrkB-mediated PI3K/Akt signaling, leading to p-AEP T322 and inactivation of AEP. Hence, though both C189S and T322E AEP mutants can be cut by endogenous active AEP, they display no enzymatic activities. On the other hand, BDNF/TrkB pathway inhibits C/EBP β , a crucial AEP transcription factor, resulting in AEP expression suppression. Hence, R13 antagonizes AEP through both post-translational phosphorylation and transcriptional repression via activating BDNF/TrkB pathway.

Are you saying that AEP is active in the cytosol and not in lysosome? I thought AEP needed acidic PH for its activity (papers published by the group of C Watts)? I am a bit confused here.

Minor comments

Figure 5 MC3T3-E4 (osteoblast lineage cell) cultured in osteogenic medium for 4 days show upregulation of RANKL, CEBPb, pCEBPb, OPG and down regulation of AEP. Similar AEP expression is detected in cells with or without osteogenic medium. In Supp Fig3, using the same cell line, AEP expression differs a lot with or without osteogenic medium. Is there an explanation for this?

A: In Fig 5C and revised Supplementary Fig 5A, OIM elevates active AEP levels in MC3T3-E4 cells as compared to control. Since inactive AEP C189S is overexpressed, highly yielding abundant enzymatic-dead "truncate form" AEP. In order to show active fragment, we present short time exposure data. In Fig 5C, since both BDNF and 7,8-DHF repressed active AEP fragmentation, in order to present active AEP fragment, we presented long time exposure results, so that the active AEP form appears comparable under control and OIM in these two Figs.

The authors did not answer the question. In fig 5C top panel Control-DMSO OIM treated MC3T3_E4 cells compared to Supp Fig 5 control-OIM-control/OIM is it the DMSO effect?

Reviewer #3 (Remarks to the Author):

All comments have been addressed.

Reviewer #4 (Remarks to the Author):

The authors have addressed the comments adequately

Response to referees

The point-by-point response to the referee #1's comments is listed as follows (the new questions are in RED, and our responses are in BLUE):

Reviewer #1 (Remarks to the Author):

A: In our previous paper, we demonstrate that Akt phosphorylates AEP on T322 and inhibits its autocleavage and blocks AEP activation. AEP is a cysteine protease that shreds the substrates after asparagine residue and it has no stringent peptide sequence in the substrates. However, it preferentially cuts the substrates at NK sites with neutral or positive charge at P-1 or -2 positions. Because the autocleavage site locates at N323, which is right next to the phosphorylation residue by Akt, the negative charge from phosphate on p-T322 prevents AEP cysteine protease autocleavage. MC3T3 cells possess high basal active AEP levels and subchronic BDNF or 7,8-DHF treatment (4 days) robustly represses active AEP formation. As requested, we present that p-AEP T322 signals associated with the upstream p-Akt/Akt levels in the revised Fig 5C.

Q: But in Fig 5C, we do not see more AEP inactive form in cells treated with BDNF or 7,8-DHF.

A: We have shown before that BDNF reduction triggers activation of C/EBP β , a crucial transcription factor for AEP, leading to AEP level escalation. On the other hand, 7,8-DHF treatment decreases active AEP levels (Wang et al., 2019, Cell Report, Fig 1; Chen et al., 2018, PNAS, Fig 6). Hence, BDNF or 7,8-DHF treatment should decrease AEP total protein levels via repressing its transcription by C/EBP β . Presumably, because "MC3T3 cells possess high basal active AEP levels", **BDNF/7,8-DHF treatment may inhibit its auto-cleavage and blunt its proteolytic activation, which leads to upregulation of inactive full-length level, counteracting its suppressive effect on AEP transcriptive expression.** Altogether, this may explain why we do not observe more AEP inactive full-length form after BDNF or 7,8-DHF treatment.

We have published several articles regarding AEP post-translational modification to regulate its activation. For instance, we show that SRPK2, a cell cycle-mediated kinase, phosphorylates AEP on S226 residue and triggers its cytoplasmic translocation from the lysosomes and proteolytic activation to yield 36 kDa active form, getting access to its substrates⁴. Moreover, we also reported that Akt phosphorylated AEP on T322 residue and sequestered its lysosomal residency, blocking its activation⁵. BDNF or 7,8-DHF treatment activates TrkB-mediated PI3K/Akt signaling, leading to p-AEP T322 and inactivation of AEP. Hence, though both C189S and T322E AEP mutants can be cut by endogenous active AEP, they display no enzymatic activities. On the other hand, BDNF/TrkB pathway inhibits C/EBP β , a crucial AEP transcription factor, resulting in AEP expression suppression. Hence, R13 antagonizes AEP through both post-translational phosphorylation and transcriptional repression via activating BDNF/TrkB pathway.

Q: Are you saying that AEP is active in the cytosol and not in lysosome? I thought AEP needed

acidic pH for its activity (papers published by the group of C Watts)? I am a bit confused here.

A: In our 2017 Mol. Cell paper, we showed that p-S226 on AEP by SRPK2 translocates it from the lysosomes into the cytoplasm and enhances its enzymatic activity. Indeed, a naïve AEP displays its optimal enzymatic activity under acidic pH conditions, such as in the lysosomes. In the previous paper, we have provided extensive evidence supporting post-translational modification on p-S226 AEP strongly escalates its protease activity even in the cytoplasm (Fig 2G, Fig3E&3F). Moreover, we have used LE-28, a small molecular probe conjugating and labeling active AEP, demonstrating that p-AEP after phosphorylated by SRPK2 not only locates outside the lysosomes but also is labeled by LE-28 (Wang et al., 2021, Prog. Neurobiol, Fig 2). This observation strongly supports that cytosolic AEP, outside the regions of LAMP1, a marker for the lysosomes, is enzymatically active.

Minor comments

Figure 5 MC3T3-E4 (osteoblast lineage cell) cultured in osteogenic medium for 4 days show upregulation of RANKL, CEBP β , pCEBP β , OPG and down regulation of AEP. Similar AEP expression is detected in cells with or without osteogenic medium. In Supp Fig3, using the same cell line, AEP expression differs a lot with or without osteogenic medium. Is there an explanation for this?

A: In Fig 5C and revised Supplementary Fig 5A, OIM elevates active AEP levels in MC3T3-E4 cells as compared to control. Since inactive AEP C189S is overexpressed, highly yielding abundant enzymatic-dead "truncate form" AEP. In order to show active fragment, we present short time exposure data. In Fig 5C, since both BDNF and 7,8-DHF repressed active AEP fragmentation, in order to present active AEP fragment, we presented long time exposure results, so that the active AEP form appears comparable under control and OIM in these two Figs.

Q: The authors did not answer the question. In fig 5C top panel Control-DMSO OIM treated MC3T3_E4 cells compared to Supp Fig 5 control-OIM-control/OIM is it the DMSO effect?

A: We did observe robust signals in the top two immunoblotting blots with anti-C/EBP β or anti-p-C/EBP β in the Fig 5C under DMSO+OIM as compared to medium control, accordingly, active AEP was elevated on the third blot versus control (lane 1 & 2). In Supplementary Fig 5A, in two OIM controls (with or without empty vector transfection control, lane 2 & 3), active AEP (37 kDa) was slightly more abundant than control (lane 1) as well. **We do not think that this discrepancy, if there is any, about active AEP levels between these two independent experiments is significant, considering the baseline for active AEP is high in MC3T3 cells.** Conceivably, other factors in MC3T3 cells might be also implicated in upregulating AEP in addition to C/EBP β . Hence, DMSO's effect in Fig 5C is negligible. Though Fig 5C's top and 2nd blot's C/EBP β appears strongly active as compared to control, **the medium contains both OIM+DMSO for lane 2, whereas lane 1 contains medium only**, therefore, they are not comparable. Thus, we cannot postulate that DMSO plays any role here either!

REVIEWERS' COMMENTS

Reviewer #1 (Remarks to the Author):

Thank you for replying to my comments.
The paper is now fine for publication.